

The variation characteristics and possible sources of atmospheric
water-soluble ions in Beijing
**Pengfei Liu**[1, 2], **Chenglong Zhang**[1], **Yujing Mu**\*, [1], **Chengtang Liu**[1,2], **Chaoyang Xue**[1, 2], **Can**
**Ye**[1, 2], **Junfeng Liu**[1], **Yuanyuan Zhang**[1], **Hongxing Zhang**[1, 3]
[1] Research Center for Eco-Environmental Sciences, Chinese Academy of Sciences, Beijing, 100085, China
[2] University of Chinese Academy of Sciences, Beijing, 100049, China
[3] Beijing Urban Ecosystem Research Station, Beijing, 100085, China
*Correspondence to:* **Yujing Mu** (yjmu@rcees.ac.cn)
Abstract: The North China plain (NCP) including Beijing is currently suffering from severe haze
events due to high pollution level of atmospheric fine particles called $PM_{2.5}$. To mitigate the serious
pollution status, identification of the sources of $PM_{2.5}$ is urgently needed for the effective control
measures. A total of 235 daily samples of $PM_{2.5}$ were collected in Beijing through the year of 2014,
and the variation characteristics of water-soluble ions (WSIs) in the $PM_{2.5}$ were comprehensively
analyzed for recognizing their possible sources. The results indicated that the periodic emissions
from farmers' activities made evident contribution to the atmospheric WSIs in Beijing. The
unusually high ratio of $Cl^-$ to $Na^+$ in summer could be rationally explained by the prevailing
fertilization of $NH_4Cl$ for planting summer maize in the vast area of NCP. The remarkable elevation
of $Cl^-$ in winter was ascribed to coal combustion for heating by farmers. The most serious pollution
episodes in autumn were coincident with significant elevation of $Ca^{2+}$ which was ascribed to be
from harvest of the summer maize and tillage for planting the winter wheat. The mineral dust
emission from the harvest and tillage not only increased the atmospheric concentrations of the
primary pollutants, but also greatly accelerated formation of sulfate and nitrate through
heterogeneous reactions of $NO_2$ and $SO_2$ on the mineral dust. The relatively high concentration of
$K^+$ in winter and autumn further confirmed that crop straw burning made evident contribution to
atmospheric $PM_{2.5}$ in Beijing. The backward trajectories also indicated that the highest
concentrations of WSIs usually occurred in the air parcel from southwest/south regions with high
density of farmers. In addition, the values of nitrogen oxidation ratio (NOR) and the sulfur oxidation
ratio (SOR) were found to be much higher under haze days than under non-haze days, implying that
formation of sulfate and nitrate was greatly accelerated through heterogeneous or multiphase
reactions of $NO_2$ and $SO_2$ on $PM_{2.5}$.
**1. Introduction**
The North China plain (NCP) is frequently suffering from severe haze pollution in recent years
(Chan and Yao, 2008;Liang et al., 2016), which has aroused great attention from the general public
(Zhang et al., 2014;Guo et al., 2014;Huang et al., 2014;Yang et al., 2015b;Zhang et al., 2015b;Zheng
et al., 2015b;Sun et al., 2006). The severe haze pollution is mainly ascribed to elevation of fine




particulate matter, usually called $PM_{2.5}$ (Huang et al., 2014). $PM_{2.5}$ can directly reduce atmospheric
visibility by scattering or absorbing solar light (Seinfeld and Pandis, 1998;Buseck and Posfai,
1999;Cheng et al., 2006) and is harmful to human health (Finlayson-Pitts and Pitts, 2000;Nel,
2005;Poschl, 2005;Peplow, 2014).
To mitigate the serious pollution status, identification of the sources of $PM_{2.5}$ is urgently needed for
the effective control measures. Based on field measurements, positive matrix factorization (PMF)
(Yu et al., 2013;Wu et al., 2014;Huang et al., 2014), principal component analysis (PCA) (Wang et
al., 2015) and chemical mass balance (CMB) (Huang et al., 2014;Guo et al., 2012) have been widely
used for identifying the sources of $PM_{2.5}$. However, the results of the source apportionment are still
not convincing because there are large uncertainties about the indicators, dominant factors and
emission inventories used for the identification. For example, some studies suggested traffic
emissions in Beijing contributed about 15~20% to the $PM_{2.5}$ (Yu et al., 2013;Wu et al., 2014), while
only 4% of the contribution was also reported (Huang et al., 2014). Additionally, the current source
apportionment can only present gross contribution of each source classification, but there are
markedly different emissions from individual sources in the same classification. For example, due
to the strict control measures and highly efficient combustion, the emissions of pollutants from
power plants and big boilers fueled by coal must be totally different from the domestic coal stoves
on both the emission strengths and composition of pollutants. Finally, most studies about source
apportionment mainly focused on emissions from traffic, industry, construction and secondary
formation, whereas the emissions from farmers' activities in the NCP were almost neglected.
There are about 300,000 $km^2$ agricultural fields and 0.16 billion farmers in the NCP (Zhang et al.,
2011). The farmers' activities in the NCP are very seasonal, e.g., the fertilization events and harvests



mainly focus on June-July and October-November and domestic coal stoves are prevailingly used
for heating in winter. The seasonal activities of farmers in the NCP were suspected to make
significant contribution to deteriorate the regional air quality, e.g., the most serious pollution events
(or haze days) in the NCP were usually coincident with the three seasonal activities of farmers in
recent years (Yang et al., 2015b;Huang et al., 2012;Li et al., 2014;Li et al., 2011;Liu et al., 2013;Sun
et al., 2013). The serious pollution events during harvest seasons were widely ascribed to crop straw
burning (Huang et al., 2012;Li et al., 2014), but the influence of fertilization events and crop straw
returning to fields on the regional air quality during the harvest seasons periods was totally neglected.
Strong ammonia ($NH_3$) emission from the vast agricultural fields in the NCP has been found during
fertilization events just after harvest of winter wheat in June-July (Zhang et al., 2011), which must
accelerate atmospheric ammonium formation. Although crop straws burning by stealth is still
prevailing, most residual crops are being returned into the agricultural fields under the advocacy of
government for protecting the air quality. Because crop leaves absorbed large quantities of
atmospheric particles during crop growing season, the abrupt release of the particles by smashing
crop straw for returning in the vast area of the NCP must also make striking contribution to
atmospheric particles in the region during the seasonal harvest seasons. In winter, the serious
pollutant emissions from the chimney of the farmers' coal stoves can be easily imagined by the
strong smog. Although domestic coal consumption only accounts for small fraction of the total, e.g.,
~11% in Beijing-Tianjin-Hebei area (http://hbdczx.mep.gov.cn/pub/), the emission strengths of
pollutants from farmers' coal stove is usually about 1-2 magnitude greater than those from power
plants (Xu et al., 2006), and the farmers coal consumption mainly concentrates on the four months
in winter.



In this study, to understand the possible influence of farmers' activities on the regional air quality in
the NCP, filter samples of $PM_{2.5}$ were daily collected in Beijing city for a whole year of 2014, and
the seasonal variation characteristics of the water-soluble ions (WSIs) in the $PM_{2.5}$ samples were
comprehensively investigated in relation to the farmers' activities. The scientific evidences found
in this study will be helpful for future control measures in reducing pollutant emissions from rural
areas in the NCP.
**2. Materials and methods**
**2.1. Sampling**
The sampling site was chosen on a rooftop (about 25m above ground) in the Research Center for
Eco-Environmental Sciences (RCEES), which is located between the north fourth-ring road and the
north fifth-ring road of Beijing and surrounded by some institutes, campuses, and residential areas
(Pang and Mu, 2006). $PM_{2.5}$ samples were collected on Millipore PTFE filters (90mm) by an
artificial intelligence's $PM_{2.5}$ sampler (LaoYing-2034) and the sampling flow rate was set to 100L
$min^{-1}$. The duration of each sampling was 24 hours, started at 3:00 p.m. every day and ended at 3:00
p.m. on the next day. All the samples were put in dedicated filter storage containers (90mm,
Millipore) after sampling and preserved in a refrigerator till analysis. A total of 235 $PM_{2.5}$ samples
were collected from January to November of 2014, in winter (Jan 9- Mar 15 2014), spring (Mar 16-
May 31 2014), summer (Jun 1- Jun 30, Aug 9- Aug 21 2014) and autumn (Sep 19- Nov 14 2014).
**2.2. Ion analysis**
Sample and blank filters were extracted ultrasonically with 10mL ultrapure water for half an hour.
The solutions were filtered through water micro-porous membrane (pore size, 0.45μm; diameter,
13mm) before analysis and the water-soluble ions (WSIs) in the treated filtrates were analyzed by





Ion Chromatography (IC, WAYEE IC6200). Five anions ($F^-$, $HCOO^-$, $Cl^-$, $NO_3^-$ and $SO_4^{2-}$) were
separated by using an anion column (IC SI-52 4E, 4mmID*250mm) with the eluent (3.6mmol $L^{-1}$
$Na_2CO_3$) flow rate of 0.8mL $min^{-1}$ and column temperature of 45 ℃. Five cations ($Na^+$, $NH_4^+$, $Mg^{2+}$,
$Ca^{2+}$ and $K^+$) were separated by using a cation column (TSKgelSuperIC-CR, 4.6mmID*15cm) with
the eluent (2.2mmol $L^{-1}$ MSA and 1mmol $L^{-1}$ 18-crown-6) flow rate of 0.7mL $min^{-1}$ and column
temperature of 40 ℃. The relative standard deviation (RSD) of each ion was less than 0.5% for the
reproducibility test. The detection limits (S/N=3) were less than 0.001 mg $L^{-1}$ for the anions and
cations. The concentrations of all the ions (less than 0.03 mg $L^{-1}$ for each ion) in daily field blank
filter were subtracted from sample determination.
**2.3. Meteorology, trace gases and back trajectory**
The meteorological data, including temperature, wind speed, wind direction, relative humidity (RH),
visibility and Air Pollution Index of $PM_{2.5}$, $SO_2$, $NO_2$, $O_3$ in RCEES were both collected from
Beijing urban ecosystem research station (http://www.bjurban.rcees.cas.cn/ ).
To identify the potential influence of air parcel transport, the air mass backward trajectories were
calculated for 72h through the Hybrid Single-Particle Lagrangian Integrated Trajectory (HYSPLIT
4) Model of the Air Resources Laboratory of NOAA with NCEP Final analyses data. The backward
trajectories arriving at 500m above sampling position were computed at 0:00h, 6:00h, 12:00h and
18:00h (UTC) each sampling day respectively. A total of 940 backward trajectories with 72 hourly
trajectory endpoints in four seasons were used as input for further analysis.
**2.4 The TEOM 1405 Monitor**
The mass concentration of $PM_{2.5}$ was monitored by a tapered element oscillating microbalance with
the filter dynamic measurement system (TEOM-FDMS, Thermo; Model 1405). A filter in the




TEOM 1405 Monitor is used for collecting and measuring $PM_{2.5}$ through variation of the oscillation
frequency. To avoid water condensation on the TEOM filter, the temperature of the TEOM filter as
well as the inlet is kept at 50 ℃ during sampling. In this study, we replaced the TEOM filters every
12 days, and the concentrations of the WSIs on the TEOM filters were analyzed for comparing with
those on the filter collected by the filter sampling method.
**3. Results and discussion**
**3.1. Comparison between WSIs and $PM_{2.5}$**
The mass concentrations of WSIs and $PM_{2.5}$ at the sampling site were simultaneously measured by
the filter sampling method and the TEOM 1405 Monitor for 24 days (Jan 1- Jan 24, 2015). As
shown in Fig. 1a and Fig. 1b, the variation trends of the WSIs and $PM_{2.5}$ were almost the same
with a correlation coefficient ($R^2$) of 0.908, implying that the concentration of WSIs measured
could well reveal the pollution status of $PM_{2.5}$ in Beijing. The average mass concentration of WSIs
contributed about 80% to the mass of $PM_{2.5}$ measured by the TEOM 1405 Monitor, whereas the
WSIs accounted for about 50-60% of the total mass concentration measured by the filter sampling
method in the NCP (Shen et al., 2009;Li et al., 2013). The mass concentration of $PM_{2.5}$ measured
by the TEOM 1405 Monitor was suspected to be largely underestimated because the volatile even
semi-volatile component in $PM_{2.5}$ can be easily lost at 50 ℃ which is designed in the TEOM 1405
Monitor for avoiding water condensation on the filter (Grover et al., 2005;Liu et al., 2014), e.g.,
under clean days after serious pollution episodes, the mass concentration of WSIs was even higher
than the mass concentration of $PM_{2.5}$ measured by the TEOM 1405 Monitor (Fig. 1a). To verify
above assumption, the concentrations of WSIs on the filters collected by the filter sampling
method and the TEOM 1405 Monitor were comparatively measured, and the results are illustrated





146 in Fig. 1c and Fig. 1d. It is evident that the proportions of $NH_4^+$, $NO_3^-$ and $Cl^-$ on the filter

147 collected by the TEOM 1405 Monitor were dramatically lower than those on the filter collected by

148 the filter sampling method, whereas $SO_4^{2-}$ was on the contrary. It is well documented that

149 temperature is a key factor affecting the distribution of both $NH_4NO_3$ and $NH_4Cl$ on particle phase

150 due to their thermo decomposition, e.g., at temperature greater than 35 °C, little $NH_4NO_3$ is

151 expected under typical ambient conditions (Finlayson-Pitts et al., 1986). The negative $PM_{2.5}$

152 values of the TEOM 1405 Monitor after a serious pollution episode also indicated the serious loss

153 of the volatile component. Although the TEOM 1405 Monitor is widely used for measuring

154 atmospheric $PM_{2.5}$ in the net stations of China, the pollution levels measured could only represent

155 the lower limits, especially under the clean days after serious pollution episodes in winter.

156 **3.2. Daily variations of WSIs in each season**

157 The daily variations of WSIs in each season are illustrated in Fig. 2 and the statistic mass

158 concentrations of the WSIs are summarized in Table 1. It is evident that the daily variations of the

159 WSIs exhibited significantly periodic fluctuation, indicating meteorological conditions played a

160 pivotal role in accumulation and dissipation of atmospheric pollutants. For example, the most

161 frequently high pollution levels of the WSIs in winter were mainly ascribed to the relatively stable

162 meteorological conditions with the low height of boundary layer which favors pollutants

163 accumulation (Wang et al., 2013;Quan et al., 2014;Tian et al., 2014;Wang et al., 2014;Zhang et al.,

164 2015a). Besides meteorological conditions, the extremely high levels of the WSIs during the

165 pollution episodes revealed strong sources of the pollutants around Beijing.

166 The mean concentrations ($\mu g\ m^{-3}$) of WSIs in spring, summer, autumn and winter were 50.5 ±37.3,

167 44.2 ±28.9, 78.3 ±92.6, and 78.7 ±61.2, respectively. $NO_3^-$, $SO_4^{2-}$ and $NH_4^+$ were found to be the





principal ions, accounted for about 80% to the total WSIs in each season, which were in line with
previous studies (Hu et al., 2014;Yang et al., 2015a;Huang et al., 2016;Yang et al., 2015b). The three
principal ions were mainly ascribed to secondary formation as discussed in the following section.
Although the most intensive photochemical reactivity in summer favors sulfate and nitrate
formation, the relatively low $SO_2$ concentration, the fast thermal decomposition of ammonium
nitrate and the frequent scavenging by rain events must greatly counteract the contribution of the
secondary formation, resulting in the lowest pollution levels of the WSIs in summer. In comparison
with other seasons, the remarkable elevation of atmospheric $SO_2$ and $NO_x$ (see Sect. 3.3) in winter
would override the relatively low atmospheric photo-oxidants for their oxidation rates and resulted
in the highest mean concentration of WSIs. Although the atmospheric concentrations of $SO_2$ and
$NO_x$ in autumn were much smaller than in winter and in spring (see Sect. 3.3), the mean
concentration of WSIs in autumn was almost the same as that in winter and nearly twice as those in
spring and summer, indicating that special mechanisms dominated the secondary formation of the
atmospheric principal ions (see Sect. 3.3).
**3.3. The possible sources for the WSIs**
To disclose the contribution of possible sources to the WSIs, the molar composition of the WSIs,
the seasonal variation characteristics of typical WSIs, the variation characteristics of the three
principal ions during serious pollution episodes, the contribution of secondary formation to
atmospheric WSIs, and backward trajectories of air parcels were comprehensively analyzed.
3.3.1. The molar composition of the WSIs
The molar composition of water-soluble ions in each season under three pollution levels is illustrated
in Fig. 3. With increasing pollution levels, the noticeable reduction of the proportions of metallic



ions (such as $Ca^{2+}$, $Na^+$ and $Mg^{2+}$) and the evident increase of $NH_4^+$, $NO_3^-$ and $SO_4^{2-}$ proportions
revealed that the three principle ions ($NH_4^+$, $NO_3^-$ and $SO_4^{2-}$) were mainly from atmospheric
secondary formation. Compared with $SO_4^{2-}$, the fast increase of $NO_3^-$ proportion with increasing
pollution levels indicated that the formation rate of nitrate was faster than that of sulfate under higher
pollution levels. It should be mentioned that the increase rate of $NO_3^-$ proportion with increasing
pollution levels was much slower in summer than in other seasons, validating that nitrate was easily
thermal decomposed under high temperature. The conspicuous reduction of $Cl^-$ proportion with
increasing pollution levels meant $Cl^-$ might be mainly from primary sources.
3.3.2. The seasonal variation characteristics of typical WSIs
The seasonal variation characteristics of typical WSIs are illustrated in Fig. 4. For $Cl^-$ and $K^+$, their
high concentrations mainly occurred in winter and autumn. It should be mentioned that the
extremely high concentration of $K^+$ in winter on 1 February (Fig. 2) was due to firework for
celebrating Chinese lunar year (Jiang et al., 2015;Kong et al., 2015). Sea-salt has long been
considered as the source for atmospheric $Cl^-$ (Souza et al., 2014), however, the molar ratio of $Cl^-$ to
$Na^+$ measured by this study (Fig. 5) in each season was above 1.30 which was much greater than
the value of 1.18 in fresh sea-salt particles (Brewer, 1975), indicating sources other than sea-salt
dominated atmospheric $Cl^-$ in Beijing. Because $K^+$ has been widely used as an indicator for biomass
burning (Gao et al., 2011) and crop straw burning by stealth was prevailing in the countryside around
Beijing during autumn and winter seasons, crop straw burning was suspected to be a common source
for $K^+$ and $Cl^-$ (Li et al., 2014). The pronounced correlation coefficients ($r > 0.6$, $p < 0.01$) between
$K^+$ and $Cl^-$ in the two seasons might be the circumstantial evidence for above suspicion. Several
studies have reported extremely high emission factors of $Cl^-$ (80-300mg $Cl^-$/kg coal) from the coal



combustion in China (Huang et al., 2014). Because large fraction of coal consumed by farmers for
heating in winter was the extra source for atmospheric pollutants in the vast area of North China,
the obviously higher $Cl^-$ concentrations measured in winter than in other seasons (Fig. 2) indicated
that coal combustion by farmers in winter might make great contribution to atmospheric $Cl^-$ in
Beijing. The source of atmospheric $NO_x$ in Beijing is dominated by vehicles and relatively stable in
the four seasons, and hence the ratios of $Cl^-$ to $NO_x$ can largely counteract the influence of
accumulation and dispersion due to variation of meteorological factors for identifying the possible
extra source of $Cl^-$. The ratio of $Cl^-$ to $NO_x$ in winter was about a factor of 2 greater than those in
other seasons (Fig. 5), confirming that coal combustion by farmers in winter indeed made evident
contribution to atmospheric $Cl^-$ in Beijing. Previous field investigations in different areas of Chinese
mainland also found relatively high $Cl^-$ concentration in winter, which was also ascribed to coal
combustion (Yu et al., 2013;Wu et al., 2014). In addition, fertilization events in the agricultural
fields around Beijing might also make contribution to atmospheric $Cl^-$, because the volatile
ammonium chloride is a kind of prevailingly used fertilizer in the NCP, e.g., the extremely high
ratios of $Cl^-$ to $Na^+$ (Fig. 5) were coincident with the cultivation seasons of spring and summer.
For $Ca^{2+}$, remarkably high concentrations occurred in both spring and autumn. The evident elevation
of $Ca^{2+}$ concentrations in spring has been usually ascribed to the frequent dust storm (Zhao et al.,
2013b), but there was still no explanation about the extremely high $Ca^{2+}$ concentrations in autumn
(Zhao et al., 2013b;Zhang et al., 2013). The three serious pollution events with remarkable elevation
of $Ca^{2+}$ (Fig. 2) were coincident with the intensive harvest of maize and tillage of the agricultural
fields for planting winter wheat in the countryside around Beijing, and hence the extremely high
$Ca^{2+}$ concentrations in autumn were suspected to be from the farmers' activities. Because abundant





atmospheric mineral particles were absorbed by crop leaves during crop growing season, especially
in the North China where atmospheric mineral dust is always at high level (Zhang et al., 2013;Zhao
et al., 2013b), a large fraction of the mineral dust absorbed on the leaves of crop would be released
into the atmosphere during harvest with crop straw being crushed into pieces for returning to fields
which is a prevailing cultivation manner under the advocacy of governments for reducing the
influence of crop straw burning on the air quality.
For $NH_4^+$, $SO_4^{2-}$ and $NO_3^-$, remarkably high concentrations also appeared in both winter and autumn.
$NH_4^+$ was mainly from the reactions of $NH_3$ with acid gases (such as $HNO_3$) and acid particles, and
hence its variation trend was the same as those of $SO_4^{2-}$ and $NO_3^-$. Although atmospheric $NH_3$ has
long been considered to be mainly from agricultural activities, their emissions mainly focus on
warmer seasons (Krupa, 2003). However, the frequently high concentrations of $NH_4^+$ appeared in
winter. Beside the slow thermal decomposition of ammonium nitrate, strong $NH_3$ emission sources
other than agricultural activities were suspected to be responsible for the frequently high
concentrations of $NH_4^+$ in the cold winter. Emissions of $NH_3$ from vehicles was regarded as an
important source (Liu et al., 2014). In addition, strong emission of $NH_3$ from domestic coal stoves
was indeed found by our preliminary measurements (data were not shown). During the serious
pollution episodes, the concentrations of $SO_2$ in autumn were almost the same as those in summer
and about one magnitude lower than in winter (Fig. 6), but the peak concentrations of $SO_4^{2-}$ in
autumn were about two times greater than those in summer and at almost the same level as those in
winter. The gaseous phase reaction with OH (Zhao et al., 2013c;Quan et al., 2014), the
heterogeneous reaction on mineral dust (He et al., 2014;Nie et al., 2014), and multiphase reactions
in the water of particulate matters (Zheng et al., 2015a) of $SO_2$ have been recognized to be





responsible for atmospheric $SO_4^{2-}$ formation. The significant elevation of both $Ca^{2+}$ and $SO_4^{2-}$ in
autumn implied that the heterogeneous reaction of $SO_2$ on the mineral dust might greatly accelerate
the conversion of $SO_2$ to $SO_4^{2-}$. Although evidently high concentrations of $Ca^{2+}$ occurred (Fig. 2 and
Fig. 4) in spring and $SO_2$ concentrations were much greater in spring than in autumn (Fig. 6), the
$SO_4^{2-}$ concentrations were about a factor of 2 less in spring than in autumn. Atmospheric humidity
was suspected to play an important role in the heterogeneous reaction, e.g., the relative humidity
was much higher in autumn than in spring during the serious pollution events (Fig. 6). Similar to
$SO_4^{2-}$, the relatively high concentrations of $NO_3^-$ during the serious pollution events in autumn were
also ascribed to the heterogeneous reaction of $NO_2$ on the mineral dust.
3.3.3. The variation characteristics of the three principal ions during serious pollution episodes
As shown in Fig. 6, the serious pollution episodes with noticeable elevation of various pollutants
usually occurred under slow wind speed (less than 2 m s$^{-1}$) and high relative humidity. In comparison
with their precursors of $SO_2$ and $NO_x$, however the detailed variation trends of $SO_4^{2-}$ and $NO_3^-$ were
different, indicating that the elevation of $SO_4^{2-}$ and $NO_3^-$ was not simply ascribed to the physical
process of accumulation. It is interesting to be noted that the increasing rates of $SO_4^{2-}$ during some
serious pollution events especially with elevation of $Ca^{2+}$ (such as in spring and autumn) were much
slower than those of $NO_3^-$, implying that the atmospheric heterogeneous reaction of $NO_2$ on the
mineral dust might be faster than that of $SO_2$. In comparison with summer and winter, the relatively
high ratios of $NO_3^-/SO_4^{2-}$ in spring and autumn (Fig. 5) also supported above assumption.
3.3.4. Secondary formation for atmospheric sulfate and nitrate
The nitrogen oxidation ratio NOR = $nNO_3^-$ / ($nNO_3^-$ + $nNO_x$) (n refers to molar concentration) and
the sulfur oxidation ratio SOR = $nSO_4^{2-}$ / ($nSO_4^{2-}$ + $nSO_2$) have been used to estimate the degree of





secondary formation of $NO_3^-$ and $SO_4^{2-}$, which can counteract the interference of meteorological
factors (Chan and Yao, 2008;Yu et al., 2013;Guo et al., 2014;Huang et al., 2014;Yang et al.,
2015b;Zheng et al., 2015b). The values of NOR and SOR during haze days and non-haze days in
four seasons are listed in Table 2. Both the values of NOR and SOR on non-haze days were found
to be the highest in summer and the lowest in winter, well reflecting the seasonal variation of
photochemical intensity. Although sunlight intensity greatly reduced at ground level during haze
days, the values of NOR and SOR were about a factor of 2 greater during haze days than during
non-haze days in the four seasons, implying again that the heterogeneous or multiphase reactions of
$SO_2$ and $NO_2$ on atmospheric particles made significant contribution to atmospheric sulfate and
nitrate.
3.3.5. The influence of air mass transport on the WSIs in Beijing
To reveal the air mass transport influence on the WSIs in Beijing, three-day backward trajectories
for clusters and the corresponding mass concentrations of WSIs in each season were analyzed, and
the results are illustrated in Fig. 7. It could be seen that the lowest concentrations of WSIs usually
occurred in the northwest/northeast airflow with long distance transport. Because Beijing is
surrounded by mountains in the north/northwest/northeast directions where the population is sparse,
these clusters brought the relatively clean air mass to accelerate the dissipation of aerosols. The
highest concentrations of WSIs (especially for $SO_4^{2-}$, $NO_3^-$ and $NH_4^+$) were usually observed in the
air parcel from southwest/south regions with high density of population. Considering the large
fraction (~30%) of air parcel from the southwest/south regions in each season, the human activities
in the southwest/south regions made evident contribution to the atmospheric WSIs in Beijing.
Besides the industries, the emissions from the high density of farmers in the southwest/south regions



of Beijing was also suspected to make evident contribution to the atmospheric WSIs in Beijing, e.g.,
the remarkable elevations of $Cl^-$ in winter and $Ca^{2+}$ in autumn were probably from farmers' coal
combustion for heating and harvest of maize, respectively.
**3.4. Comparison with previous studies**
The mean concentrations of the three principal ions and some related indicators in Beijing over the
past decade are summarized in Table 3. The seasonal variations of the three principal ions reported
were quite different, e.g., Huang et al. (2016) found the maximal mean concentrations of $SO_4^{2-}$ and
$NH_4^+$ in the summer and of $NO_3^-$ in the autumn of 2014, whereas in this study all the maximal mean
concentrations of the three principal ions appeared in autumn. The mean concentrations of the three
ions in autumn in this study were in good agreement with the values reported by Yang et al. (2015).
For the mass concentration ratios of $NO_3^-/SO_4^{2-}$ (denoted as N/S), all the investigations exhibited
relatively high values in autumn and spring, further confirming that the heterogeneous reaction of
$NO_2$ on mineral dust favored nitrate formation (as discussed above). For NOR and SOR, all
investigations were in good agreement, with the highest values in summer, the lowest in winter and
higher values during haze days than during clean days. Compared with the investigations of 2003,
the evident increase of both the concentration of $NO_3^-$ and the ratio of N/S in recent years revealed
the fast increase of vehicle numbers in the decade made significant contribution to atmospheric
nitrate.
**4. Conclusions**
The comparison between the mass concentrations of WSIs measured by the filter method and the
mass concentrations of $PM_{2.5}$ measured by the TEOM 1405 Monitor revealed that the mass
concentrations of WSIs could well reflect the pollution status of $PM_{2.5}$ and the mass concentrations



of $PM_{2.5}$ measured by the TEOM 1405 Monitor were evidently underestimated due to the serious
loss of volatile components in the atmospheric particulate matters.
The conspicuous daily fluctuation of the WSIs in each season confirmed that meteorological factors
played an important role in governing the accumulation and dispersion of the pollutants. The
extremely high concentrations of the WSIs during the serious pollution episodes indicated there
were strong sources of the pollutants in Beijing. Based on the comprehensive analysis of the data of
the WSIs, the strongly periodic activities of farmers, such as crop harvest, crop straw burning, and
coal combustion for heating, were found to make evident contribution to the atmospheric WSIs in
Beijing. To mitigate the currently serious pollution status in the NCP including Beijing, the strong
emissions of pollutants from the periodic activities of farmers should be aroused great attention.
**Author contribution**
**Y. J. Mu** designed the experiments and prepared the manuscript. **P. F. Liu** carried out the
experiments and prepared the manuscript. **C. L. Zhang** carried out the experiments. **C. T. Liu**, **C.**
**Y. Xue**, **C. Ye**, **J. F. Liu** and **Y. Y. Zhang** were involved in part of the work. **H. X. Zhang** provided
the meteorological data and trace gases.
**Acknowledgements**
This work was supported by the National Natural Science Foundation of China (21477142,
41203070 and 91544211), the "Strategic Priority Research Program" of the Chinese Academy of
Sciences (XDB05010100) and the Special Fund for Environmental Research in the Public Interest

341    (201509002).

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



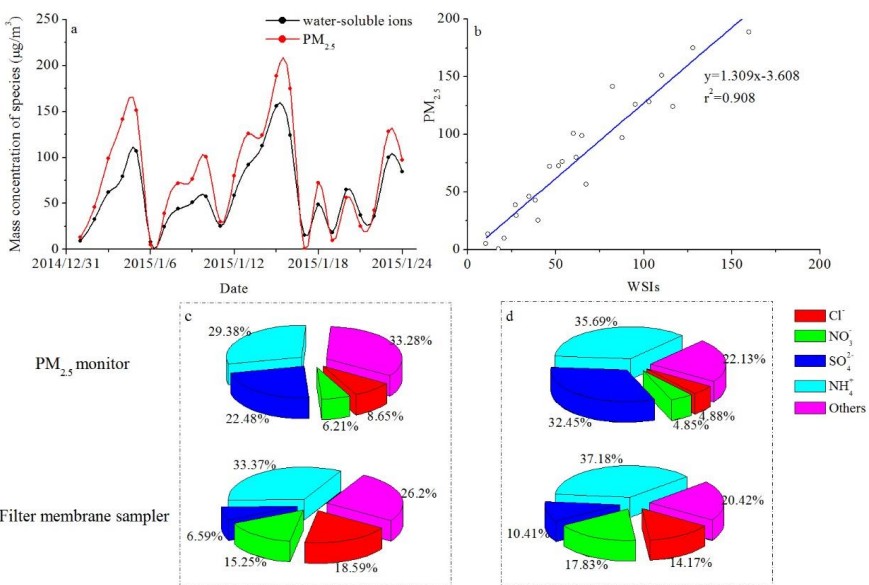


**Fig. 1** Comparison between the filter sampling method and the PM$_{2.5}$ monitor for the daily average mass
concentrations of the WSIs and PM$_{2.5}$ (Fig. 1a and 1b), and for the 12-day-average molar composition of the WSIs
on the filters collected by the two methods during the two 12-day sampling periods (Fig. 1c represents the data
collected during the first 12-day; Fig. 1d represents the data collected during the second 12-day.).

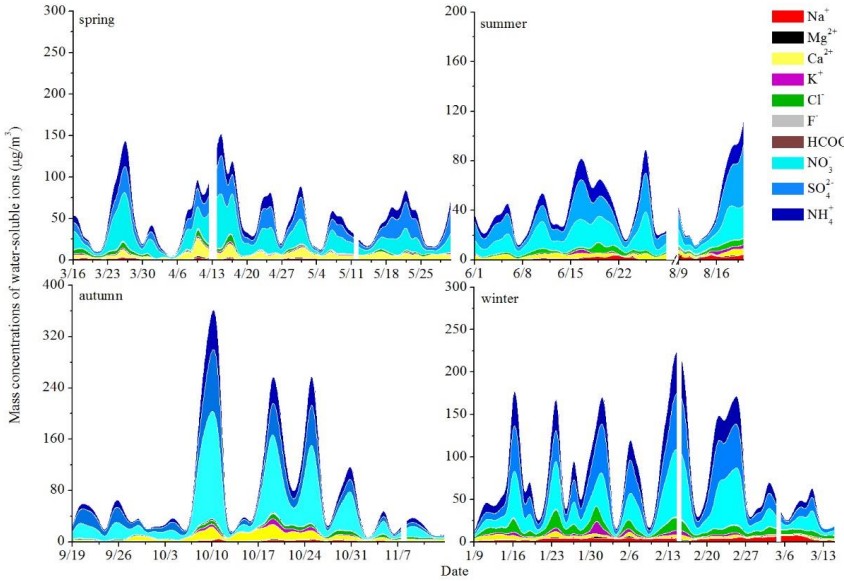


**Fig. 2** Daily variations of WSIs in each season (the smooth lines for the WSIs were drawn between the points of
the daily data).




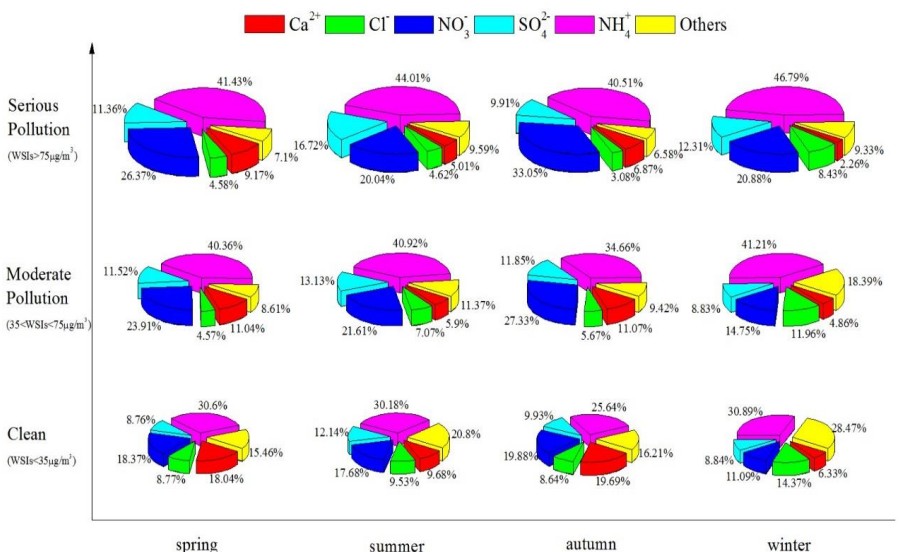

**Fig. 3** Molar composition of the WSIs under different pollution levels in four seasons (Clean: WSIs < 35μg m$^{-3}$;
Moderate pollution: 35μg m$^{-3}$ < WSIs < 75μg m$^{-3}$; Serious pollution: WSIs > 75μg m$^{-3}$)

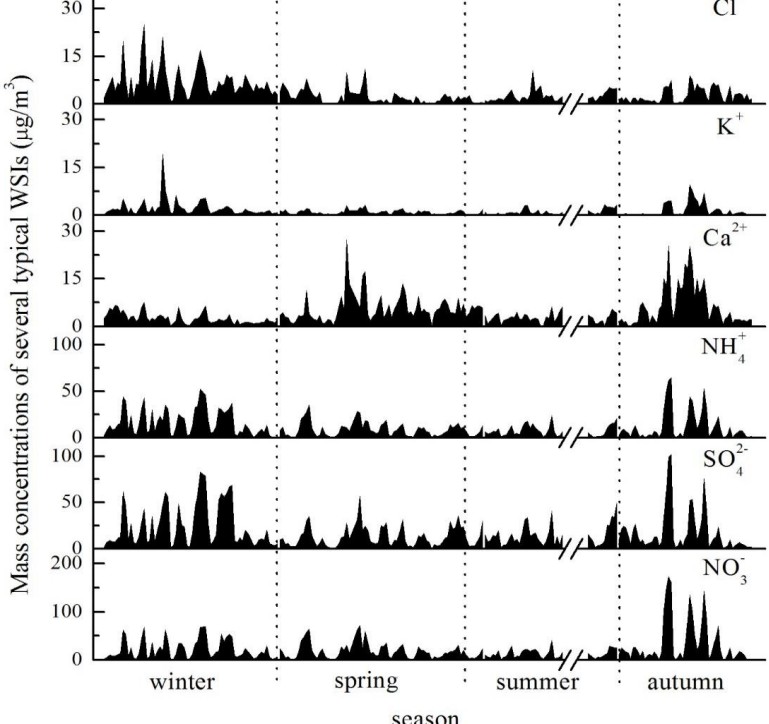

**Fig. 4** The seasonal variations of the several typical WSIs in the year of 2014





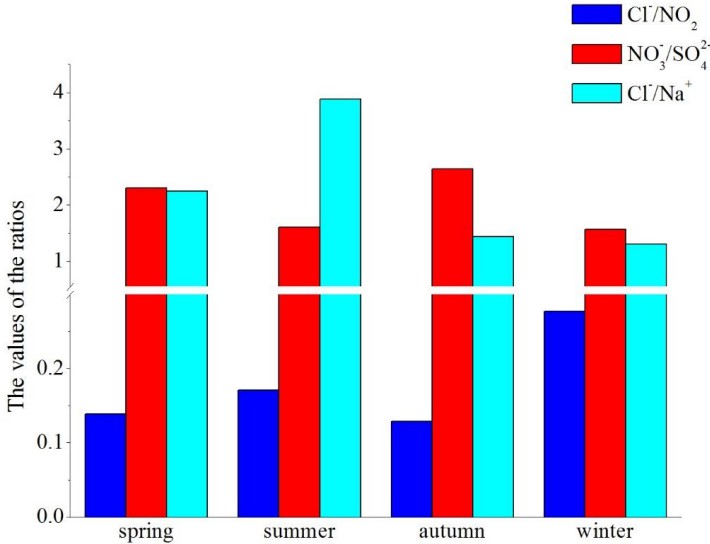


**Fig. 5** the average ratio of Cl$^-$/NO$_2$ (the unit is μg/m$^3$ and ppb, respectively) and the average molar ratios of Cl$^-$/Na$^+$ and NO$_3^-$/SO$_4^{2-}$ in each season.

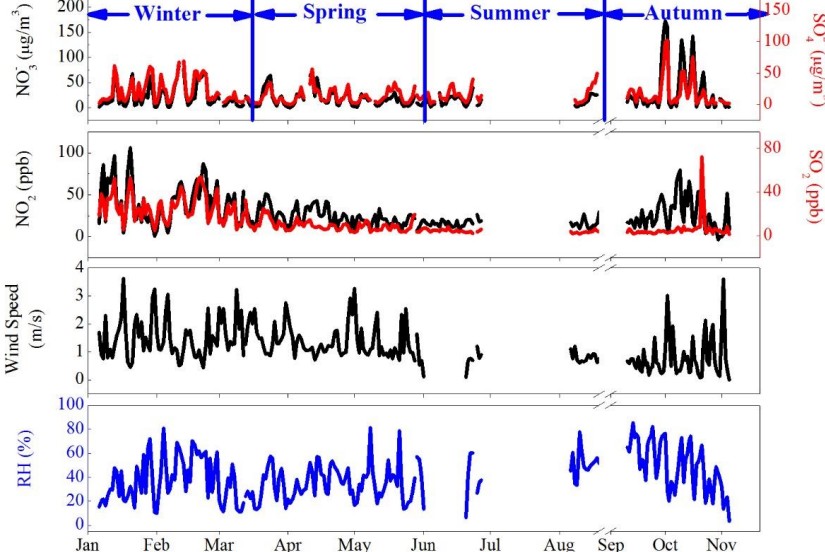


**Fig. 6** Time series of NO$_3^-$, SO$_4^{2-}$, NO$_2$ and SO$_2$ and meteorological data (wind speed and relative humidity) in four seasons for 2014







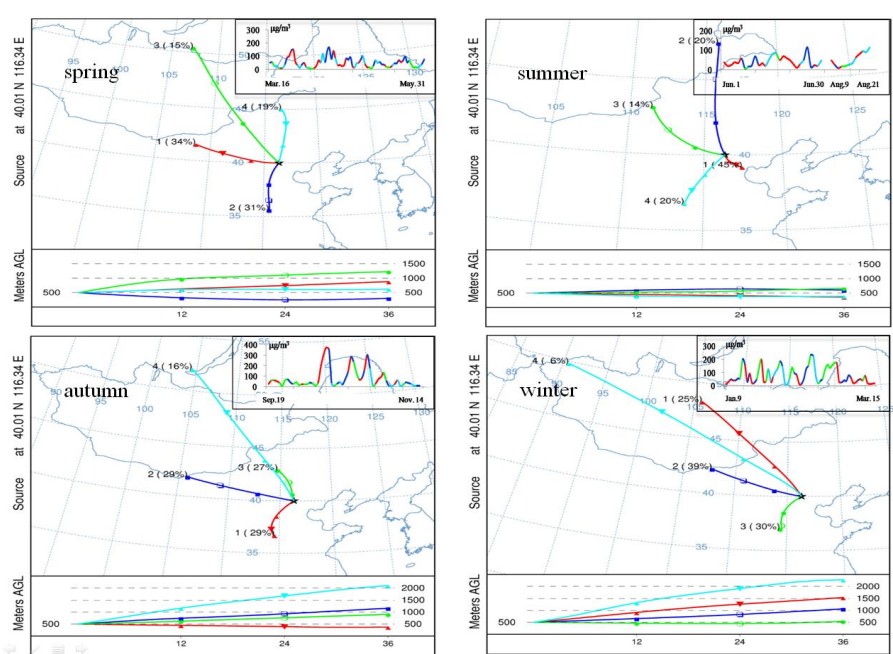

**Fig. 7** The back trajectory cluster analysis and the corresponding overall ion mass concentration in four seasons

**Table 1** Concentrations (μg m$^{-3}$) of the WSIs (mean concentrations and standard deviation (SD)) in four seasons in Beijing.

| Species | Spring (N=74) | | Summer (N=41) | | Autumn (N=56) | | Winter (N=64) | | Annual (N=235) | |
|---|---|---|---|---|---|---|---|---|---|---|
| | Mean | SD | Mean | SD | Mean | SD | Mean | SD | Mean | SD |
| F$^-$ | 0.3 | 0.3 | 0.2 | 0.1 | 0.4 | 0.2 | 0.2 | 0.2 | 0.3 | 0.2 |
| HCOO$^-$ | 0.2 | 0.1 | 0.2 | 0.1 | 0.4 | 0.5 | 0.3 | 0.2 | 0.3 | 0.3 |
| Cl$^-$ | 2.4 | 2.2 | 2.6 | 1.9 | 2.8 | 2.3 | 7.0 | 4.9 | 3.9 | 3.7 |
| NO$_3^-$ | 18.4 | 16.0 | 13.4 | 9.3 | 34.3 | 45.2 | 23.8 | 22.8 | 22.8 | 27.7 |
| SO$_4^{2-}$ | 13.0 | 10.9 | 14.6 | 11.6 | 18.1 | 22.8 | 22.2 | 19.6 | 17.0 | 17.3 |
| Na$^+$ | 1.2 | 0.8 | 2.1 | 1.4 | 1.6 | 1.1 | 3.8 | 1.7 | 2.3 | 1.8 |
| NH$_4^+$ | 8.8 | 7.4 | 7.6 | 6.0 | 12.3 | 16.3 | 16.5 | 13.6 | 11.5 | 12.2 |
| Mg$^{2+}$ | 0.5 | 0.4 | 0.3 | 0.2 | 0.4 | 0.3 | 0.5 | 0.5 | 0.4 | 0.4 |
| Ca$^{2+}$ | 5.6 | 4.2 | 2.9 | 1.5 | 6.8 | 6.4 | 2.6 | 1.8 | 4.6 | 4.4 |
| K$^+$ | 1.0 | 0.7 | 1.1 | 1.0 | 1.6 | 2.2 | 2.2 | 2.7 | 1.5 | 1.9 |
| Mass | 50.5 | 37.3 | 44.2 | 28.9 | 78.3 | 92.6 | 78.7 | 61.2 | 63.7 | 62.0 |





577      **Table 2** SOR and NOR during haze days and non-haze days in four seasons.

|  | Spring | | Summer | | Autumn | | Winter | |
|---|---|---|---|---|---|---|---|---|
|  | SOR | NOR | SOR | NOR | SOR | NOR | SOR | NOR |
| Haze days | 0.3 | 0.3 | 0.7 | 0.4 | 0.6 | 0.4 | 0.2 | 0.3 |
| Non-haze days | 0.2 | 0.2 | 0.3 | 0.2 | 0.3 | 0.2 | 0.1 | 0.1 |
| Ratio | 1.8 | 1.8 | 2.0 | 2.3 | 2.0 | 2.6 | 2.3 | 2.5 |

578      The ratio of values in Haze days to that in Non-haze days.



**Table 3** Summary of three principal ions ($\mu g\ m^{-3}$), the mass concentration ratio of $NO_3^-/SO_4^{2-}$ (denoted as N/S), NOR and SOR for four seasons in Beijing.

| Year | Spring $NO_3^-$ | $SO_4^{2-}$ | $NH_4^+$ | N/S | NOR | SOR | Summer $NO_3^-$ | $SO_4^{2-}$ | $NH_4^+$ | N/S | NOR | SOR | Autumn $NO_3^-$ | $SO_4^{2-}$ | $NH_4^+$ | N/S | NOR | SOR | Winter $NO_3^-$ | $SO_4^{2-}$ | $NH_4^+$ | N/S | NOR | SOR | Reference |
|---|---|---|---|---|---|---|---|---|---|---|---|---|---|---|---|---|---|---|---|---|---|---|---|---|---|
| 2014 | 18.4 | 13.0 | 8.8 | 1.4 | 0.2 | 0.2 | 13.4 | 14.6 | 7.6 | 0.9 | 0.2 | 0.4 | 34.3 | 18.1 | 12.3 | 1.9 | 0.2 | 0.4 | 23.8 | 22.2 | 16.5 | 1.1 | 0.2 | 0.2 | This work |
| 2014(haze) | 30.2 | 21.6 | 14.5 | 1.4 | 0.3 | 0.3 | 25.0 | 28.8 | 15.3 | 0.9 | 0.4 | 0.7 | 73.6 | 36.0 | 26.5 | 2.0 | 0.4 | 0.6 | 37.7 | 34.5 | 25.4 | 1.1 | 0.3 | 0.2 | This work |
| 2014(clean) | 7.8 | 5.2 | 3.5 | 1.5 | 0.2 | 0.2 | 8.6 | 8.7 | 4.4 | 1.0 | 0.2 | 0.3 | 8.9 | 6.5 | 3.2 | 1.4 | 0.2 | 0.3 | 5.9 | 6.4 | 4.5 | 0.9 | 0.1 | 0.1 | This work |
| 2014 | - | - | - | - | - | - | 35.5 | - | - | - | - | - | 35.5 | 20.0 | 16.7 | 1.8 | 0.2 | 0.4 | - | - | - | - | - | - | Yang et al., 2015b |
| 2013-2014(haze) | 14.7 | 9.0 | 10.3 | 1.6 | 0.2 | 0.4 | 33.9 | 32.7 | 24.0 | 1.0 | 0.3 | 0.7 | 40.0 | 17.4 | 22.2 | 2.3 | 0.2 | 0.6 | 22.0 | 20.4 | 18.8 | 1.1 | 0.2 | 0.2 | Huang et al., 2016 |
| 2013-2014(clean) | 3.6 | 2.4 | 4.4 | 1.5 | 0.1 | 0.2 | 8.8 | 8.1 | 11.7 | 1.1 | 0.1 | 0.4 | 5.5 | 4.5 | 5.6 | 1.2 | 0.1 | 0.4 | 6.6 | 5.2 | 6.0 | 1.3 | 0.1 | 0.1 | Huang et al., 2016 |
| 2013(haze) | - | - | - | - | - | - | - | - | - | - | - | - | - | - | - | - | - | - | 26.1 | 33.3 | 24.1 | 0.8 | - | - | Tian et al., 2014 |
| 2013(clean) | - | - | - | - | - | - | - | - | - | - | - | - | - | - | - | - | - | - | 4.9 | 5.0 | 4.9 | 1.0 | - | - | Tian et al., 2014 |
| 2010(haze) | - | - | - | - | - | - | - | - | - | - | - | - | - | - | - | - | - | - | - | - | - | - | 0.5 | 0.3 | Zhao et al., 2013a |
| 2010(clean) | - | - | - | - | - | - | - | - | - | - | - | - | - | - | - | - | - | - | - | - | - | - | 0.3 | 0.2 | Zhao et al., 2013a |
| 2009-2010 | 15.5 | 14.7 | 7.5 | 1.1 | - | - | 11.8 | 23.5 | 11.0 | 0.5 | - | - | 10.7 | 7.9 | 4.7 | 1.4 | - | - | 7.3 | 8.5 | 4.5 | 0.9 | - | - | Zhang et al., 2013 |
| 2009 | - | - | - | - | - | - | 12.7 | 26.1 | 9.1 | 0.5 | 0.2 | 0.7 | 6.1 | 20.1 | 4.3 | 0.3 | 0.1 | 0.6 | - | - | - | - | - | - | Hu et al., 2014 |
| 2005 | - | - | - | - | - | - | 9.9 | 22.6 | 4.7 | 0.4 | - | - | - | - | - | - | - | - | - | - | - | - | - | - | Pathak et al., 2009 |
| 2001-2003 | 11.9 | 13.5 | 6.5 | 0.9 | 0.1 | 0.1 | 11.2 | 18.4 | 10.1 | 0.6 | 0.1 | 0.4 | 9.1 | 12.7 | 6.3 | 0.7 | 0.1 | 0.2 | 12.3 | 21.0 | 10.6 | 0.6 | 0.1 | 0.1 | Wang et al., 2005 |
| 2002-2003 | - | - | - | - | - | - | 12.2 | 16.0 | 10.4 | 0.8 | - | - | - | - | - | - | - | - | 17.0 | 30.4 | 12.9 | 0.6 | - | - | Sun et al., 2004 |