# Peer review of "Published: 16 March 2016"

_Atmospheric Chemistry and Physics, 2016_

## Referee Comment (RC1) · Anonymous Referee #1 · 24 Mar 2016

In this study, over two hundred PM2.5 samples were collected in four seasons, and only the water-soluble ions were analysed. So many studies have already been carried out for the chemical compositions from PM2.5 in Beijing. Because of the lack of other related aerosol measurements, this paper basically focuses on the simple display of the ion concentrations. Most of the discussion are based on speculation, and no new ideas and no interesting points are found in this paper. On the whole, this paper is not suitable for publication in the ACP. In addition, there are also some problems and mistakes in this paper. After major revision, this paper might be suitable for publication in some local journals. 1. It is strongly recommended that this paper be send to a language editing service. There are too many Chinese English in this paper. For example,

the use of the word "farmer" is inaccurate, even ridiculous, just as "with high density of famers", "farmers' activities", "heating by farmers". At present, most of the people living in the rural area are not engaged in agricultural activities. And farmers have also not engaged in agricultural activities in most of the time. You should use the "rural area" and "agricultural activities" to describe the exact meaning. 2. Line 70, "Because crop leaves absorbed large quantities of atmospheric particles during crop growing season, the abrupt release of the particles by smashing crop straw for returning in the vast area of the NCP must also make striking contribution to atmospheric particles in the region during the seasonal harvest seasons." This statement is basically impossible to be true. There is no evidence that the crop could absorb PM2.5. And the smashing process of crop straw could not be an important source of PM2.5. Just a small amount coarse PM might be emitted. 3. Line 74, what's the meaning of "pollutant emissions from the chimney of the farmers' coal stoves"? There is not a thing called "farmers' coal stoves" in this world. I think "pollutants from coal combustion for heating" is more accurate. The author is not familiar with the countryside. 4. Line 94, "dedicated filter storage containers"? I think it should be a desiccator. 5. As mentioned in this paper, the TEOM 1405 is not suitable for accurate PM2.5 mass concentration measurement owing to the volatilization of unstable components. Why didn't the authors weigh the PTFE filters before and after the sampling for mass concentration analysis? This is the biggest problem in this paper. The proportions of different ions in PM2.5 could not be obtained.

———————————————————————

---

## Author Comment (AC1) · 27 Mar 2016

Thank you for your comments. But we don't totally agree with your opinions. The followings are our responses to your comments.

**Comment 1:** So many studies have already been carried out for the chemical compositions from $PM_{2.5}$ in Beijing. Because of the lack of other related aerosol measurements, this paper basically focuses on the simple display of the ion concentrations. Most of the discussion are based on speculation, and no new ideas and no interesting points are found in this paper. (the words marked in red are incorrect in English)

**Answer:** Although many studies have already been carried out for the chemical composition in $PM_{2.5}$ in Beijing, most of them focused on summer or winter, and only 5 of them conducted measurements in the four seasons. In addition, all of the studies only conducted measurements less than one month in one season. To reflect the seasonal variation characteristics of the water-soluble ions in detail, the daily measurements were conducted in this study except for July.
Although we didn't conduct other related aerosol measurements, the variation characteristics of the composition of the water-soluble ions in $PM_{2.5}$ were found to well reflect their possible sources, and the following important conclusions were derived from the measurements: 1. With the elevation of $Ca^{2+}$ in spring and autumn, the evidently faster increasing rates of $NO_3^-$ than $SO_4^{2-}$ implied that the atmospheric heterogeneous reaction of $NO_2$ on the mineral dust was an important source for $NO_3^-$; 2. The obviously higher $Cl^-$ concentrations and the remarkably greater ratios of $Cl^-$ to $NO_x$ in winter than in other seasons indicated that coal combustion by farmers in winter made great contribution to atmospheric $Cl^-$ in Beijing; 3. The extremely high ratios of $Cl^-$ to $Na^+$ in summer implied that fertilization with ammonium chloride in the agricultural fields around Beijing might make evident contribution to atmospheric $Cl^-$; 4. The coincidence of the remarkable elevation of $Ca^{2+}$ with the intensive harvest of maize and tillage of the agricultural fields in the vast rural areas around Beijing revealed that the harvest and tillage made striking contribution to atmospheric $Ca^{2+}$ in Beijing. Because the typical ratios of atmospheric pollutants have been widely used for identifying the sources of the pollutants, the above conclusions based on the remarkable variation characteristics of typical ratios were not simply from speculation. To our best knowledge, there are still no reports about the above conclusions which will be helpful for future control measures in reducing pollutant emissions from rural areas in the North China Plain. Additionally, the heterogeneous reaction of $NO_2$ on mineral dusts has been found to make contribution to nitrate formation under laboratory simulations, but the role of the reaction for nitrate formation has not been recognized in field measurements before this study.

**Comment 2:** On the whole, this paper is not suitable for publication in the ACP.

**Answer:** Because field measurement is one of the main subject areas of the ACP and there are original findings (comment 1) in the paper, we wondered why you say that the paper is not suitable for publication in the ACP.

**Comment 3:** In addition, there are also some problems and mistakes in this paper. After major

revision, this paper might be suitable for publication in some local journals. It is strongly recommended that this paper be send to a language editing service. There are too many Chinese English in this paper. For example, the use of the word "farmer" is inaccurate, even ridiculous, just as "with high density of famers", "farmers' activities", "heating by farmers". At present, most of the people living in the rural area are not engaged in agricultural activities. And **farmers** have also not engaged in agricultural activities in most of the time. You should use the "rural area" and "agricultural activities" to describe the exact meaning.

**Answer:** "Farmer" is a commonly used word to represent people who are living in rural areas (Pattey et al., *Journal of the Air & Waste Management Association*, 2012:62(7); Mahmud, *Geofizika*, 2009:26(1)), why did you say "the use of the word "farmer" is inaccurate, even ridiculous"? We don't think it is polite to use the word "ridiculous" in your comment. We don't understand your meaning about the description of "agricultural activities" in the two sentences. Because the "rural area" was used for representing the countryside in the text of the paper, why did you request us using the "rural area" to describe the exact meaning? The "farmers' activities" in this paper included both "agricultural activities" and farmers' living activities (cooking and heating via coal combustion, etc.), and hence "farmers' activities" is more exact than "agricultural activities" for describing our meaning.

**Comment 4:** Line 70, "Because crop leaves absorbed large quantities of atmospheric particles during crop growing season, the abrupt release of the particles by smashing crop straw for returning in the vast area of the NCP must also make striking contribution to atmospheric particles in the region during the seasonal harvest seasons." This statement is basically impossible to be true. There is no evidence that the crop could absorb PM2.5. And the smashing process of crop straw could not be an important source of PM2.5. Just a small amount coarse PM might be emitted.

**Answer:** We don't think the reviewer know that plants play an important role in PMs (including $PM_{10}$ and $PM_{2.5}$) uptake (Bealey et al., *Journal of Environmental Management*, 2007:85(1); Ji et al., *Science China*, 2013:43(8)). There are about 300,000 km$^2$ agricultural fields where the harvest of winter wheat or summer maize only lasts about two weeks in the North China Plain, and hence one can imagine the huge emission of mineral dust during the harvest in the photo of the attachment. The evident elevation of $Ca^{2+}$ in $PM_{2.5}$ was also found during the summer maize harvest season in the rural area and Beijing city in 2014 (the figure in the attachment).

**Comment 5:** Line 74, what's the meaning of "pollutant emissions from the chimney of the farmers' coal stoves"? There is not a thing called "farmers' coal stoves" in this world. I think "pollutants from coal combustion for heating" is more accurate. The author is not familiar with the countryside.

**Answer:** We don't think "pollutants from coal combustion for heating" is more accurate for describing the meaning of the sentence, because "pollutants from coal combustion for heating" includes varies sources from industrial boilers, central-heating boilers as well as domestic coal stoves. In our opinions, it is better to replace "the farmers' coal stoves" with "the domestic coal

stoves". The corresponding author of this paper was born and grew up in a village of the North China Plain and frequently visits the village every year. In addition, our group has been engaged in field measurements of $N_2O$ emissions for about ten years. Therefore, we are familiar with the rural areas very well.

**Comment 6:** Line 94, "dedicated filter storage containers"? I think it should be a desiccator.

**Answer:** The dedicated filter storage container is not a desiccator but a kind of dedicated box for storing the filters. The objective of this paper is to investigate the water-soluble ions in $PM_{2.5}$ not to measure the mass concentrations of $PM_{2.5}$, and hence desiccators were not used as containers for the filters. The dedicated filter storage containers are commercial products which have been widely used for storing the filters by investigators.

**Comment 7:** As mentioned in this paper, the TEOM 1405 is not suitable for accurate PM2.5 mass concentration measurement owing to the volatilization of unstable components. Why didn't the authors weigh the PTFE filters before and after the sampling for mass concentration analysis? This is the biggest problem in this paper. The proportions of different ions in PM2.5 could not be obtained.

**Answer:** Because we lack the precision balance for weighing the filters, the mass concentrations of $PM_{2.5}$ were not measured. The mass concentrations of $PM_{2.5}$ can reveal the pollution levels, but not the detail information about their sources' origination. In this paper, the variation characteristics of the water-soluble ions provided the important information about the evident contribution of farmers' activities and heterogeneous reaction of $NO_2$ on mineral dust to the components of atmospheric $PM_{2.5}$ in Beijing. Why did you say "this is the biggest problem in this paper"? We have measured all of the ions in $PM_{2.5}$, why do you conclude that "the proportions of different ions in $PM_{2.5}$ could not be obtained"?

---

## Short Comment (SC1) · 7 Apr 2016

The manuscript presents results from ion chromatography analysis of samples of PM2.5 collected in Beijing through the year of 2014 aimed at deriving the variation characteristics of water-soluble ions (WSIs) in the PM2.5. Since only a small part of studies focused on the variation characteristics of WSIs in the four seasons by now, I think the intentions of the authors are very good and substantial data about the WSIs in the PM2.5 are provided which can make an incremental gain in the knowledge of the haze occurred in Beijing. The science is sound and the results are meaningful. In addition, the authors are very familiar with the North China plain (NCP) and the agriculture activities and living activities of farmers in NCP. There are

interesting findings that the emissions from farmers' activities in the NCP was one possible emission sources and the influence of fertilization events and crop straw have influence on the regional air quality during the harvest seasons periods which have been neglected by most previous studies. Maybe more attention would be paid to the agriculture activities in NCP and that some field observations would be carried out in rural area after this paper. The detailed data of the daily variations of WSIs in this paper showed an obvious seasonal variation characteristic, which may helpful for further exploring how meteorological factor affect the accumulation and dispersion of atmospheric pollutants. And it was found that the atmospheric concentrations of $SO_2$ and $NO_2$ in autumn are much smaller than that in winter and spring, whereas the mean concentration of WSIs in autumn was almost the same as that in winter and nearly twice as that in spring. This result indicates that unknown mechanisms of atmospheric heterogeneous reactions and transformation of atmospheric pollutants from gas phase to particulate phase should be investigated. Moreover, it was an interesting observation that the increasing rates of $SO_4^{2-}$ during some serious pollution events were much slower than those of $NO_3^-$, especially with the elevation of $Ca^{2+}$. The heterogeneous reactions of $SO_2$ and $NO_2$ with mineral dust may be an important pathway for the formation of sulfate and nitrate in the urban cities of East Asia because of the frequent occurrence of dust storms. Most previous studies focused on the heterogeneous uptake of $SO_2$ or $NO_2$ on mineral aerosol surfaces without considering coexistent gases in atmospheric condition. Only a few studies reported that $SO_2$ and $NO_2$ likely exert synergistic effect on the surface of mineral dust. To my knowledge there is still a lack of knowledge to explain why the increase of nitrate proportion with increasing pollution levels much faster than the increase of sulfate. I'm interested in the new ideas and inspiring points in this paper.

Please also note the supplement to this comment:
http://www.atmos-chem-phys-discuss.net/acp-2016-82/acp-2016-82-SC1-supplement.pdf

---

## Author Comment (AC2) · 10 Apr 2016

Dear Professor Maofa Ge, Thank you for your positive view of our study. As you've said, the pollutant emissions from the agriculture activities and living activities of farmers in the North China Plain (NCP) have been indeed neglected by most previous studies. We have carried out the related field measurements in the rural area of the NCP and the preliminary experimental results could also confirm the conclusions in this paper very well. The further analysis about the comparison between the rural area and the urban region for PM2.5 will be discussed in other papers.

Thank you for your helpful comments.

[Figure]

Sincerely, Yujing Mu and Pengfei Liu

---

## Referee Comment (RC2) · Anonymous Referee #2 · 11 May 2016

In this study Liu et al. characterized the water-soluble ions (WSIs) of PM2.5 in Beijing on the basis of one-year filter sampling. This kind of intensive field and lab experiment is laborious. However, the authors fail to provide new findings and/or sound conclusion that can advance our understanding of haze pollution in Beijing, compared with previous dozens of publications. Most of important, solid evidence is critically needed to support their statement regarding the possible sources from agricultural activities. In addition, the concentrations of WSIs are so high that close to PM2.5 (e.g., Figure 1), arising the concern of the data quality. In general, organics are equally as important as WSIs in PM2.5, especially during days with lower PM2.5. If possible, the authors should perform mass closure studies to ensure the data quality.

[Figure]

Detailed evidence to support the points is critically needed. For example: (a) Line 15: Farmers' activities; (b) Line 17: fertilization of NH4Cl; (c) Line 18: Cl- from coal combustion by farmers.

Line 18: Biomass/biofuel burning also contributes to Cl- emissions in winter?

Line 19: Mineral dust, including Ca, was transported from farmland to urban region? Construction activities also contributed to high values of Ca in urban region.

Line 27, Note that industrial emissions from south regions in NCP are also massive.

Line 36: PM2.5 is not defined due to haze. The terminology should be clarified.

Line 47-48: The authors should specify what traffic emissions included here, particles, gas, or both? Is it true that 4% of PM2.5 was attributed to vehicle exhaust from Huang et al., 2014? This may be a wrong citation.

Line 55: How does this work advance our knowledge?

Line 65: totally?

Line 73: What is the size of the particle on the crop leaves? More information is needed to show how long it can be transported. Also, wind speed is a key factor here.

Line 80: The authors should provide more solid evidence to show farmers' influences on an urban site in BJ?

Line 84: Can the contribution be quantified in this study?

Line 91: Is this kind of filter suitable for the sampling at the site with high loading of PM2.5?

Line 93: Why started at 3 pm, background information is needed.

Line 99: How blank filters are sampled? It is better to show the blank values.

Line 114: How far is it from the observation site? Are the meteorological data and air

pollutants similar at these two different sites?

Line 116: Why 72h and 500m above sampling position were selected?

Line 128: Are there new findings by using this filter sampling method, compared with method described in section 2.1?

Line 135: How the mass of PM2.5 filter was determined?

Line 140: Base on the comparison between filter sampling method and the TEOM 1405 Monitor, the authors can give out the underestimated percentage of concentrations of PM2.5 and WSIs due to the volatile even semi-volatile component.

Line 154: This may be a good point to argue, but more details are needed.

Line 192: Why nitrate was faster than sulfate under higher pollution levels.

Line 195: Please show the pattern in different seasons.

Line 213-215: Detail explanation was needed here, how can the authors identify that coal combustion by farmers in winter might make great contribution to atmospheric Cl- other than coal combustion from urban area?

Line 217: Why the ratio Cl- to NOx was selected? They are different in phases in the atmosphere.

Line 229-233: Again, more direct evidences are needed, if the authors wish to link the Ca in urban site to farmland.

Line 248-249: This is an important point and the evidence is critically needed.

Figure 1: The concentrations of WSIs are so high that close to PM2.5. In general, organics are also as equally important as WSIs in Beijing, especially during days with lower PM2.5. Mass closure studies are needed to check the data quality.

---

## Author Comment (AC3) · 23 Jun 2016

A point-by-point response to the reviews

Thank you for your valuable comments. The followings are our responses to your comments.

**Response to Reviewer #2**
**Comment 1:** In this study Liu et al. characterized the water-soluble ions (WSIs) of $PM_{2.5}$ in Beijing on the basis of one-year filter sampling. This kind of intensive field and lab experiment is laborious. However, the authors fail to provide new findings and/or sound conclusion that can advance our understanding of haze pollution in Beijing, compared with previous dozens of publications. Most of important, solid evidence is critically needed to support their statement regarding the possible sources from agricultural activities.

**Answer:** Thank you for your valuable comments. To support our statements, the typical WSIs ($Cl^-$, $Ca^{2+}$ and $K^+$) from a rural site (Fig. R1) are also presented in Fig. R2 to reveal the impact of periodic activities of farmers on the atmospheric WSIs. The rural site is far away from cities and industries, and thus the variation characteristics of atmospheric WSIs in the rural site are mainly affected by periodic farmers' activities and meteorological factors. Compared with the sampling site in Beijing city where coal has been almost replaced with natural gas and electricity for heating before 2013 (http://www.radiotj.com/gnwyw/system/2014/07/22/000485853.shtml ), the extremely high concentrations of $Cl^-$ in the rural site in winter indicated residential coal combustion for heating made evident contribution to atmospheric $Cl^-$; the obviously high concentrations of $Cl^-$ in the rural site during the basal fertilization period for maize in June implied that volatilization of the prevailing $NH_4Cl$ fertilizer under high temperature was an important source for atmospheric $Cl^-$; the relatively high concentrations of $Ca^{2+}$ in June and October were ascribed to wheat harvest and maize harvest followed by soil ploughing, respectively; the obvious elevation of $K^+$, $Cl^-$ and $Ca^{2+}$ in the rural site in November when straw burning was prevailing in the region demonstrated their strong emissions from straw burning. To recognize the impact of the periodic emissions from farmers' activities on atmospheric WSIs in Beijing, the molar proportions of atmospheric WSIs in Beijing were comparatively analyzed before, during and after the periods of heating in winter, maize fertilization in summer, and maize harvest and soil ploughing in autumn (Fig. R3). Because the atmospheric $Cl^-$ sources from sea-salt, industries, power plants and biofuels are relatively stable during the whole year and the average mass $Cl^-/K^+$ ratio of 7.1 (except for firework event during the Spring Festival) in winter was about a factor of 2 greater than the value of 3.8 in autumn when straw burning was prevailing in the region, the obvious elevation of $Cl^-$ proportion (Fig. R3) as well as $Cl^-$ concentrations (Fig. R2) in winter should be ascribed to the additional source of residential coal combustion. Besides $Cl^-$, the serious emissions of various pollutants from residential coal combustion (Zhang and Tao, 2008; Zhang et al., 2008; Li et al., 2016) must make evident contribution to deteriorate the air quality in Beijing during the wintertime. Compared with the periods before and after maize fertilization, the proportion of $Cl^-$ during maize fertilization in summer increased about 3%-4%, confirming the influence of maize fertilization on atmospheric $Cl^-$ in Beijing. Because fertilization is an important source for atmospheric $NH_3$, the elevation of $Cl^-$ (as a tracer for fertilization) revealed that fertilization in the rural areas around Beijing could also make obvious contribution to atmospheric $NH_4^+$ in Beijing. The remarkable elevation of $Ca^{2+}$ proportion in Beijing during the period of the maize harvest and

soil ploughing provided convincing evidences that the agricultural activities indeed influenced on atmospheric $Ca^{2+}$ in Beijing. The above discussion has been added in our revised manuscript.

**Comment 2:** In addition, the concentrations of WSIs are so high that close to $PM_{2.5}$ (e.g., Figure 1), arising the concern of the data quality. In general, organics are equally as important as WSIs in $PM_{2.5}$, especially during days with lower $PM_{2.5}$. If possible, the authors should perform mass closure studies to ensure the data quality.

**Answer:** The comparison between the WSIs and $PM_{2.5}$ measured by the TEOM monitor is far from the topic of the manuscript, and hence this part has been delated in our revised manuscript. According to your valuable suggestions, we will perform mass closure studies in the near future.

**Comment 3:** Detailed evidence to support the points is critically needed. For example: (a) Line 15: Farmers' activities; (b) Line 17: fertilization of $NH_4Cl$; (c) Line 18: $Cl^-$ from coal combustion by farmers.

**Answer:** The evident elevation of $Cl^-$ and $K^+$ in Beijing during the autumn indicated biomass burning, one of the farmers' activities, was an important source for atmospheric WSIs, which was in good agreement with previous studies (Wang et al., 2005; Souza et al., 2014; Yang et al., 2016). The proportion of $Cl^-$ was much higher during basal fertilization for maize in summer than before and after the fertilization event (Fig. R3) and the extremely high ratio of $Cl^-$ to $Na^+$ in summer among the four seasons well revealed the contribution of volatilization of the prevailing $NH_4Cl$ fertilizer (Ishikawa et al., 2015). The distinct seasonal variation of $Cl^-$ (Fig. R2), the proportion of $Cl^-$ in WSIs (Fig. R3) and the ratio of $Cl^-$ to $K^+$ could reflect the contribution of coal combustion by farmers to atmospheric $Cl^-$.

**Comment 4:** Line 18: Biomass/biofuel burning also contributes to $Cl^-$ emissions in winter?

**Answer:** Yes, biomass and biofuel burning could also contribute to $Cl^-$ emissions in winter (Christian et al., 2010; Li et al., 2014). However, the emission of biofuel burning is relatively stable during the whole year and the average mass $Cl^-/K^+$ ratio of 7.1 (except for firework event during the Spring Festival) in winter was about a factor of 2 greater than the value of 3.8 in autumn when biomass (straw) burning was prevailing in the region. Therefore, the obvious elevation of $Cl^-$ proportion in WSIs as well as the extremely high $Cl^-$ concentrations in winter should be ascribed to the additional source of residential coal combustion.

**Comment 5:** Line 19: Mineral dust, including $Ca^{2+}$, was transported from farmland to urban region? Construction activities also contributed to high values of $Ca^{2+}$ in urban region.

**Answer:** Yes, construction activities are an important source for atmospheric $Ca^{2+}$ in urban region. However, there are few construction activities in the rural area, which couldn't explain the extremely high concentrations of $Ca^{2+}$ over there during the autumn (Fig. R2). The extremely high concentrations of $Ca^{2+}$ in Beijing occurred during the period of 6-25 October when the air parcels were mainly from the southwest/south regions (Fig. R4) where the vast areas of agricultural field

were under intensive maize harvest and soil ploughing. Although the concentrations of $Ca^{2+}$ in the rural area were still kept high levels during the period of 2-14 November (Fig. R2), the relatively low concentrations of $Ca^{2+}$ in Beijing were observed during the period when the air parcels were mainly from the northwest region (Fig. R4) where agricultural activities are relatively sparse. Considering the relatively stable contribution of construction activities to mineral dust during each season (Zhu et al., 2005), the coincident elevation of $Ca^{2+}$ in both the rural and urban areas and the evident increase of $Ca^{2+}$ proportion in WSIs of Beijing during the period of 6-25 October (Fig. R2 and Fig. R3) revealed the influence of the maize harvest and soil ploughing in the rural area on atmospheric $Ca^{2+}$ in Beijing.

**Comment 6:** Line 27, Note that industrial emissions from south regions in NCP are also massive.

**Answer:** There are massive industrial emissions from south regions in NCP. However, the emission of industries is relatively stable during the whole year (Gao et al., 2014), which cannot explain the distinct variations of the molar proportions of atmospheric WSIs in Beijing before, during and after the periods of heating in winter, maize fertilization in summer, and maize harvest and soil ploughing in autumn.

**Comment 7:** Line 36: $PM_{2.5}$ is not defined due to haze. The terminology should be clarified.

**Answer:** The mistake has been corrected in our revised manuscript: "The severe haze pollution is mainly ascribed to elevation of fine particulate matter with dynamic diameter less than 2.5μm $(PM_{2.5})$".

**Comment 8:** Line 47-48: The authors should specify what traffic emissions included here, particles, gas, or both? Is it true that 4% of $PM_{2.5}$ was attributed to vehicle exhaust from Huang et al., 2014? This may be a wrong citation.

**Answer:** According to your valuable comments, we specify the traffic emissions. The traffic emissions reported in these references only included particles. Sorry, the reference should be Zhang et al., 2013. The mistakes have been corrected in our revised manuscript.

**Comment 9:** Line 55: How does this work advance our knowledge?

**Answer:** According to your valuable comments, the seasonal variation characteristics of WSIs in a rural site (Baoding, Hebei Province) have been added in our revised manuscript to advance our knowledge about the emissions from farmers' activities. Farmers' activities were found to make evident contribution to atmospheric WSIs in Beijing, based on the investigations about the seasonal variation characteristics of WSIs in both the rural and urban areas, and the distinct variations of the molar proportions of atmospheric WSIs in Beijing before, during and after the periods of heating in winter, maize fertilization in summer, and maize harvest and soil ploughing in autumn.

**Comment 10:** Line 65: totally?

**Answer:** "Totally" has been replaced with "mostly".

**Comment 11:** Line 73: What is the size of the particle on the crop leaves? More information is needed to show how long it can be transported. Also, wind speed is a key factor here.

**Answer:** The size of the particle on the crop leaves was not measured in this study. The previous studies confirmed that various plants can absorb atmospheric $PM_{2.5}$ and $PM_{10}$ (Bealey et al., 2007; Ji et al., 2013). There are about 300,000 $km^2$ agricultural fields where the harvest of wheat or maize mainly concentrates about two weeks in the NCP, and hence the emissions of mineral dust are suspected to be massive during the harvest through the harvest scene (Fig. R5). Although we don't know how long the particle from the harvest can be transported, the remarkable elevation of $Ca^{2+}$ proportion in Beijing during the period of the maize harvest and soil ploughing provided convincing evidences that the agricultural activities indeed influenced on atmospheric $Ca^{2+}$ in Beijing. Both wind speed and wind direction are indeed key factors for the transportation, while back trajectory is widely used for recognizing the transportation of pollutants. The extremely high concentrations of $Ca^{2+}$ in Beijing occurred during the period of 6-25 October when the air parcels were mainly from the southwest/south regions (Fig. R4) where the vast areas of agricultural field were under intensive maize harvest and soil ploughing. Although the concentrations of $Ca^{2+}$ in the rural area were still kept high levels during the period of 2-14 November (Fig. R2), the relatively low concentrations of $Ca^{2+}$ in Beijing were observed during the period when the air parcels were mainly from the northwest region (Fig. R4) where agricultural activities are relatively sparse. According to your valuable comments, we will perform the study about the size of the particle on the crop leaves in the near future.

**Comment 12:** Line 80: The authors should provide more solid evidence to show farmers' influences on an urban site in BJ?
Line 229-233: Again, more direct evidences are needed, if the authors wish to link the $Ca^{2+}$ in urban site to farmland.
Line 213-215: Detail explanation was needed here, how can the authors identify that coal combustion by farmers in winter might make great contribution to atmospheric $Cl^-$ other than coal combustion from urban area?

**Answer:** Solid evidence has been added in our revised manuscript (See the answers for comments 1, 3, 4 and 5).

**Comment 13:** Line 84: Can the contribution be quantified in this study?

**Answer:** It is difficult to quantify the contribution in this study because of the complex sources of atmospheric WSIs as well as the impact of meteorological factors. We are conducting the emission factors of various pollutants from typical farmers' activities such as residential coal combustion, the $NH_3$ emissions of agricultural field and so on, which will be helpful to quantify the contribution in the near future.

**Comment 14:** Line 91: Is this kind of filter suitable for the sampling at the site with high loading of $PM_{2.5}$?

**Answer:** The PTFE filter is widely used for PM sampling in previous studies (Chow et al., 1996; Walker et al., 2006; Pathak et al., 2009; Chen et al., 2015; Park et al., 2015). The significant correlation between WSIs sampled by the filters and $PM_{2.5}$ measured by the TEOM monitor (Fig. R6a), and the near equilibrium between cations and anions in the four seasons (Fig. R6b) indicated that this kind of filter is suitable for the sampling at the site with high loading of $PM_{2.5}$.

**Comment 15:** Line 93: Why started at 3 pm, background information is needed.
Line 99: How blank filters are sampled? It is better to show the blank values.

**Answer:** To conveniently replace the filter sample in each day, we select 3 p.m. as our starting time. Blank filters were brought to the field and were installed in the samplers which no air was pumped. After sampling, all the filters samples including blank filters were put in dedicated filter storage containers (90mm, Millipore) and preserved in a refrigerator till ion analysis. All the ion concentrations were corrected for blanks. The average blank values were about 0.03mg $L^{-1}$ for $Na^+$, $Ca^{2+}$, $F^-$, $NO_3^-$ and $SO_4^{2-}$, 0.02mg $L^{-1}$ for $NH_4^+$ and $Cl^-$, 0.01mg $L^{-1}$ for $Mg^{2+}$, $K^+$ and $HCOO^-$. According to your valuable comments, the blank values have been shown detailedly in our manuscript.

**Comment 16:** Line 114: How far is it from the observation site? Are the meteorological data and air pollutants similar at these two different sites?

**Answer:** There are about 20m between the observation station and our sampling site at almost the same height of 25m.

**Comment 17:** Line 116: Why 72h and 500m above sampling position were selected?

**Answer:** Due to the regional meteorological conditions with about 4-7 days periodic cycle (Guo et al., 2014), 72h is usually selected as the least elapsed time for recognizing regional transportation. Considering the surrounding terrain of Beijing and the height of planet boundary layer, air parcel with the height of 500m is recommended by NOAA for tracing their sources. In addition, the parameters have also been employed by previous studies (Li et al., 2012; Wang et al., 2015; Yang et al., 2016).

**Comment 18:** Line 128: Are there new findings by using this filter sampling method, compared with method described in section 2.1?
Line 135: How the mass of $PM_{2.5}$ filter was determined?
Line 140: Base on the comparison between filter sampling method and the TEOM 1405 Monitor, the authors can give out the underestimated percentage of concentrations of $PM_{2.5}$ and WSIs due to the volatile even semi-volatile component.
Line 154: This may be a good point to argue, but more details are needed.
Figure 1: The concentrations of WSIs are so high that close to $PM_{2.5}$. In general, organics are also

as equally important as WSIs in Beijing, especially during days with lower $PM_{2.5}$. Mass closure studies are needed to check the data quality.

**Answer:** As mentioned above, the comparison between the WSIs and $PM_{2.5}$ measured by the TEOM monitor is far from the topic of the manuscript, and hence this part has been delated in our revised manuscript.

**Comment 19:** Line 192: Why nitrate was faster than sulfate under higher pollution levels.
Line 195: Please show the pattern in different seasons.

**Answer:** The faster increase of nitrate proportion than that of sulfate proportion from clean days to serious pollution days mainly occurred in spring and autumn when the concentration levels of $Ca^{2+}$ were relatively high. To recognize the influence of $Ca^{2+}$ concentrations on the formation of nitrate and sulfate, the formation rates of nitrate and sulfate were analyzed under typical cases of haze formation in the four seasons (Fig. R7). It is evident that the faster formation rates of nitrate than those of sulfate only occurred under the relatively high levels of $Ca^{2+}$ in spring and autumn, indicating that the mineral dust could preferentially promote nitrate formation.

**Comment 20:** Line 217: Why the ratio $Cl^-$ to $NO_x$ was selected? They are different in phases in the atmosphere.

**Answer:** $NO_x$ in Beijing is dominated by vehicles and relatively stable during the whole year. Although $Cl^-$ and $NO_x$ are different in phases in the atmosphere, the $Cl^-/NO_x$ ratio value can counteract the influence of meteorological factors and reveal the additional sources for atmospheric $Cl^-$ in the four seasons. Considering this situation, the $Cl^-/NO_x$ ratio has been delated in our revised manuscript.

**Comment 21:** Line 248-249: This is an important point and the evidence is critically needed.

**Answer:** $NH_3$ emissions generated from a prevailing residential coal stove fueled with raw bituminous coal were investigated under alternation cycles of flaming and smoldering combustion in our preliminary studies. The $NH_3$ emission factor for the residential coal stove was recorded as 0.62-1.10g/kg coal, which was in line with Li et al., 2016. These results indicated that residential coal combustion may be a significant $NH_3$ emission source in the cold winter, and hence leading to the elevation of atmospheric $NH_4^+$ in Beijing.

**References**
Bealey, W. J., McDonald, A. G., Nernitz, E., Donovan, R., Dragosits, U., Duffy, T. R., and Fowler, D.: Estimating the reduction of urban $PM_{10}$ concentrations by trees within an environmental information system for planners, Journal of Environmental Management, 85, 44-58, 10.1016/j.jenvman.2006.07.007, 2007.
Chen, W., Tong, D., Zhang, S., Dan, M., Zhang, X., and Zhao, H.: Temporal variability of atmospheric particulate matter and chemical composition during a growing season at an agricultural site in northeastern China, J Environ Sci (China), 38, 133-141,

10.1016/j.jes.2015.05.023, 2015.

Chow, J. C., Watson, J. G., Lu, Z. Q., Lowenthal, D. H., Frazier, C. A., Solomon, P. A., Thuillier, R. H., and Magliano, K.: Descriptive analysis of $PM_{2.5}$ and $PM_{10}$ at regionally representative locations during SJVAQS/AUSPEX, Atmospheric Environment, 30, 2079-2112, 10.1016/1352-2310(95)00402-5, 1996.

Christian, T. J., Yokelson, R. J., Cardenas, B., Molina, L. T., Engling, G., and Hsu, S. C.: Trace gas and particle emissions from domestic and industrial biofuel use and garbage burning in central Mexico, Atmospheric Chemistry and Physics, 10, 565-584, 10.5194/acp-10-565-2010, 2010.

Gao, J., Tian, H., Cheng, K., Lu, L., Wang, Y., Wu, Y., Zhu, C., Liu, K., Zhou, J., Liu, X., Chen, J., and Hao, J.: Seasonal and spatial variation of trace elements in multi-size airborne particulate matters of Beijing, China: Mass concentration, enrichment characteristics, source apportionment, chemical speciation and bioavailability, Atmospheric Environment, 99, 257-265, 10.1016/j.atmosenv.2014.08.081, 2014.

Guo, S., Hu, M., Zamora, M. L., Peng, J., Shang, D., Zheng, J., Du, Z., Wu, Z., Shao, M., Zeng, L., Molina, M. J., and Zhang, R.: Elucidating severe urban haze formation in China, Proceedings of the National Academy of Sciences of the United States of America, 111, 17373-17378, 10.1073/pnas.1419604111, 2014.

Ishikawa, N., Ishioka, G., Yanaka, M., Takata, K., and Murakami, M.: Effects of Ammonium Chloride Fertilizer and its Application Stage on Cadmium Concentrations in Wheat (Triticum aestivumL.) Grain, Plant Production Science, 18, 137-145, 10.1626/pps.18.137, 2015.

Ji, J., Wang, G., Du, X., Jin, C., Yang, H., Liu, J., Yang, Q., Tchouopou Lontchi, J., Li, J., and Chang, C.: Evaluation of Adsorbing Haze $PM_{2.5}$ Fine Particulate Matters with Plants in Beijing-Tianjin-Hebei Region in Chinac, Scientia Sinica Vitae, 43, 694-699, 2013.

Li, J., Song, Y., Mao, Y., Mao, Z., Wu, Y., Li, M., Huang, X., He, Q., and Hu, M.: Chemical characteristics and source apportionment of $PM_{2.5}$ during the harvest season in eastern China's agricultural regions, Atmospheric Environment, 92, 442-448, 10.1016/j.atmosenv.2014.04.058, 2014.

Li, M. M., Huang, X., Zhu, L., Li, J. F., Song, Y., Cai, X. H., and Xie, S. D.: Analysis of the transport pathways and potential sources of $PM_{10}$ in Shanghai based on three methods, Sci. Total Environ., 414, 525-534, 10.1016/j.scitotenv.2011.10.054, 2012.

Li, Q., Jiang, J. K., Cai, S. Y., Zhou, W., Wang, S. X., Duan, L., and Hao, J. M.: Gaseous Ammonia Emissions from Coal and Biomass Combustion in Household Stoves with Different Combustion Efficiencies, Environmental Science & Technology Letters, 3, 98-103, 10.1021/acs.estlett.6b00013, 2016.

Mahmud, M.: Mesoscale equatorial wind prediction in Southeast Asia during a haze episode of 2005, Geofizika, 26, 67-84, 2009.

Park, S., Cho, S. Y., and Bae, M. S.: Source identification of water-soluble organic aerosols at a roadway site using a positive matrix factorization analysis, The Science of the total environment, 533, 410-421, 10.1016/j.scitotenv.2015.07.004, 2015.

Pathak, R. K., Wu, W. S., and Wang, T.: Summertime $PM_{2.5}$ ionic species in four major cities of China: nitrate formation in an ammonia-deficient atmosphere, Atmospheric Chemistry and Physics, 9, 1711-1722, 2009.

Pattey, E., and Qiu, G.: Trends in primary particulate matter emissions from Canadian agriculture, Journal of the Air & Waste Management Association, 62, 737-747, 10.1080/10962247.2012.672058,

2012.

Rio, M., Franco-Uria, A., Abad, E., and Roca, E.: A risk-based decision tool for the management of organic waste in agriculture and farming activities (FARMERS), Journal of Hazardous Materials, 185, 792-800, 10.1016/j.jhazmat.2010.09.090, 2011.

Souza, D. Z., Vasconcellos, P. C., Lee, H., Aurela, M., Saarnio, K., Teinila, K., and Hillamo, R.: Composition of $PM_{2.5}$ and $PM_{10}$ Collected at Urban Sites in Brazil, Aerosol and Air Quality Research, 14, 168-176, 10.4209/aaqr.2013.03.0071, 2014.

Sun, Y., Zhuang, G., Tang, A., Wang, Y., and An, Z.: Chemical Characteristics of $PM_{2.5}$ and $PM_{10}$ in Haze-Fog Episodes in Beijing, Environ. Sci. Technol., 40, 3148-3155, 2006.

Walker, J. T., Robarge, W. P., Shendrikar, A., and Kimball, H.: Inorganic $PM_{2.5}$ at a U.S. agricultural site, Environ Pollut, 139, 258-271, 10.1016/j.envpol.2005.05.019, 2006.

Wang, Y., Zhuang, G. S., Tang, A. H., Yuan, H., Sun, Y. L., Chen, S. A., and Zheng, A. H.: The ion chemistry and the source of $PM_{2.5}$ aerosol in Beijing, Atmospheric Environment, 39, 3771-3784, 10.1016/j.atmosenv.2005.03.013, 2005.

Wang, Y. H., Liu, Z. R., Zhang, J. K., Hu, B., Ji, D. S., Yu, Y. C., and Wang, Y. S.: Aerosol physicochemical properties and implications for visibility during an intense haze episode during winter in Beijing, Atmospheric Chemistry and Physics, 15, 3205-3215, 10.5194/acp-15-3205-2015, 2015.

Wang, Y., Yao, L., Wang, L., Liu, Z., Ji, D., Tang, G., Zhang, J., Sun, Y., Hu, B., and Xin, J.: Mechanism for the formation of the January 2013 heavy haze pollution episode over central and eastern China, Science China Earth Sciences, 57, 14-25, 10.1007/s11430-013-4773-4, 2013.

Xhoxhi, O., Pedersen, S. M., Lind, K. M., and Yazar, A.: The Determinants of Intermediaries' Power over Farmers' Margin-Related Activities: Evidence from Adana, Turkey, World Development, 64, 815-827, 2014.

Yang, Y., Zhou, R., Yan, Y., Yu, Y., Liu, J., Di, Y., Du, Z., and Wu, D.: Seasonal variations and size distributions of water-soluble ions of atmospheric particulate matter at Shigatse, Tibetan Plateau, Chemosphere, 145, 560-567, 10.1016/j.chemosphere.2015.11.065, 2016.

Zhang, Y., Schauer, J. J., Zhang, Y., Zeng, L., Wei, Y., Liu, Y., and Shao, M.: Characteristics of particulate carbon emissions from real-world Chinese coal combustion, Environ. Sci. Technol., 42, 5068-5073, 10.1021/es7022576, 2008.

Zhang, Y. X., and Tao, S.: Seasonal variation of polycyclic aromatic hydrocarbons (PAHs) emissions in China, Environmental Pollution, 156, 657-663, 10.1016/j.envpol.2008.06.017, 2008.

Zhao, X. J., Zhao, P. S., Xu, J., Meng, W., Pu, W. W., Dong, F., He, D., and Shi, Q. F.: Analysis of a winter regional haze event and its formation mechanism in the North China Plain, Atmospheric Chemistry and Physics, 13, 5685-5696, 10.5194/acp-13-5685-2013, 2013.

Zheng, B., Zhang, Q., Zhang, Y., He, K. B., Wang, K., Zheng, G. J., Duan, F. K., Ma, Y. L., and Kimoto, T.: Heterogeneous chemistry: a mechanism missing in current models to explain secondary inorganic aerosol formation during the January 2013 haze episode in North China, Atmospheric Chemistry and Physics, 15, 2031-2049, 10.5194/acp-15-2031-2015, 2015.

Zhu, X., Zhang, Y., Zeng, L., and Wang, W.: Source Identification of Ambient $PM_{2.5}$ in Beijing, Research of Environmental Sciences, 18, 1-5, 2005.

[Figure]

**Fig. R1** Sampling sites (the urban site in Beijing city and the rural site in Baoding, Hebei Province) in the NCP.

[Figure]

**Fig. R2** Seasonal variations of the several typical WSIs in the year of 2014. (The mass concentrations of Cl⁻, K⁺, Ca²⁺, NH₄⁺, SO₄²⁻ and NO₃⁻ were presented at RCEES and DBT. The green square showed the firework event during the period of the Spring Festival. The gray square represented farmers' activities, including residential coal combustion for heating (a), top dressing for wheat (b), wheat harvest and basal fertilization for maize (c), top dressing for maize (d), maize harvest and soil ploughing (e) and straw burning (f).)

[Figure]

**Fig. R3** Molar proportions of atmospheric WSIs in Beijing before, during and after the periods of heating in winter, maize fertilization in summer, and maize harvest and soil ploughing in autumn.

[Figure]

**Fig. R4** The back trajectory cluster analysis and the corresponding overall ion mass concentration in four seasons.

[Figure]

**Fig. R5** The harvest scene during the wheat harvest of 2014 in the rural area (close to our rural site) in Baoding, Hebei Province.

[Figure]

**Fig. R6** The correlation between WSIs sampled by the filters and PM$_{2.5}$ measured by the TEOM monitor (Fig. R6a, 1-24 January, 2015), and the ratios of cations to anions in the four seasons of 2014 (Fig. R6b).

[Figure]

**Fig. R7** Case studies about the increasing rates of NO$_3^-$ and SO$_4^{2-}$ with the elevation of Ca$^{2+}$ during serious pollution events in the four seasons.

---

## Author Response (AR1)

A point-by-point response to the reviews

Thank you for your valuable comments. The followings are our responses to your comments.

**Response to Reviewer #2**

**Comment 1:** In this study Liu et al. characterized the water-soluble ions (WSIs) of $PM_{2.5}$ in

Beijing on the basis of one-year filter sampling. This kind of intensive field and lab experiment is laborious. However, the authors fail to provide new findings and/or sound conclusion that can advance our understanding of haze pollution in Beijing, compared with previous dozens of publications. Most of important, solid evidence is critically needed to support their statement regarding the possible sources from agricultural activities.

**Answer:** Thank you for your valuable comments. To support our statements, the typical WSIs ($Cl^-$, $Ca^{2+}$ and $K^+$) from a rural site (Fig. R1) are also presented in Fig. R2 to reveal the impact of periodic activities of farmers on the atmospheric WSIs. The rural site is far away from cities and industries, and thus the variation characteristics of atmospheric WSIs in the rural site are mainly affected by periodic farmers' activities and meteorological factors. Compared with the sampling site in Beijing city where coal has been almost replaced with natural gas and electricity for heating before 2013 (http://www.radiotj.com/gnwyw/system/2014/07/22/000485853.shtml ), the extremely high concentrations of $Cl^-$ in the rural site in winter indicated residential coal combustion for heating made evident contribution to atmospheric $Cl^-$; the obviously high concentrations of $Cl^-$ in the rural site during the basal fertilization period for maize in June implied that volatilization of the prevailing $NH_4Cl$ fertilizer under high temperature was an important source for atmospheric $Cl^-$; the relatively high concentrations of $Ca^{2+}$ in June and October were ascribed to wheat harvest and maize harvest followed by soil ploughing, respectively; the obvious elevation of $K^+$, $Cl^-$ and $Ca^{2+}$ in the rural site in November when straw burning was prevailing in the region demonstrated their strong emissions from straw burning. To recognize the impact of the periodic emissions from farmers' activities on atmospheric WSIs in Beijing, the molar proportions of atmospheric WSIs in Beijing were comparatively analyzed before, during and after the periods of heating in winter, maize fertilization in summer, and maize harvest and soil ploughing in autumn (Fig. R3). Because the atmospheric $Cl^-$ sources from sea-salt, industries, power plants and biofuels are relatively stable during the whole year and the average mass $Cl^-/K^+$ ratio of 7.1

(except for firework event during the Spring Festival) in winter was about a factor of 2 greater than the value of 3.8 in autumn when straw burning was prevailing in the region, the obvious elevation of $Cl^-$ proportion (Fig. R3) as well as $Cl^-$ concentrations (Fig. R2) in winter should be ascribed to the additional source of residential coal combustion. Besides $Cl^-$, the serious emissions of various pollutants from residential coal combustion (Zhang and Tao, 2008; Zhang et al., 2008;

Li et al., 2016) must make evident contribution to deteriorate the air quality in Beijing during the wintertime. Compared with the periods before and after maize fertilization, the proportion of $Cl^-$

during maize fertilization in summer increased about 3%-4%, confirming the influence of maize fertilization on atmospheric $Cl^-$ in Beijing. Because fertilization is an important source for atmospheric $NH_3$, the elevation of $Cl^-$ (as a tracer for fertilization) revealed that fertilization in the rural areas around Beijing could also make obvious contribution to atmospheric $NH_4^+$ in Beijing.

The remarkable elevation of $Ca^{2+}$ proportion in Beijing during the period of the maize harvest and soil ploughing provided convincing evidences that the agricultural activities indeed influenced on
atmospheric $Ca^{2+}$ in Beijing. The above discussion has been added in our revised manuscript.
**Comment 2:** In addition, the concentrations of WSIs are so high that close to $PM_{2.5}$ (e.g., Figure
1), arising the concern of the data quality. In general, organics are equally as important as WSIs in
$PM_{2.5}$, especially during days with lower $PM_{2.5}$. If possible, the authors should perform mass
closure studies to ensure the data quality.
**Answer:** The comparison between the WSIs and $PM_{2.5}$ measured by the TEOM monitor is far
from the topic of the manuscript, and hence this part has been delated in our revised manuscript.
According to your valuable suggestions, we will perform mass closure studies in the near future.
**Comment 3:** Detailed evidence to support the points is critically needed. For example: (a) Line
15: Farmers' activities; (b) Line 17: fertilization of $NH_4Cl$; (c) Line 18: $Cl^-$ from coal combustion
by farmers.
**Answer:** The evident elevation of $Cl^-$ and $K^+$ in Beijing during the autumn indicated biomass
burning, one of the farmers' activities, was an important source for atmospheric WSIs, which was
in good agreement with previous studies (Wang et al., 2005; Souza et al., 2014; Yang et al., 2016).
The proportion of $Cl^-$ was much higher during basal fertilization for maize in summer than before
and after the fertilization event (Fig. R3) and the extremely high ratio of $Cl^-$ to $Na^+$ in summer
among the four seasons well revealed the contribution of volatilization of the prevailing $NH_4Cl$
fertilizer (Ishikawa et al., 2015). The distinct seasonal variation of $Cl^-$ (Fig. R2), the proportion of
$Cl^-$ in WSIs (Fig. R3) and the ratio of $Cl^-$ to $K^+$ could reflect the contribution of coal combustion
by farmers to atmospheric $Cl^-$.
**Comment 4:** Line 18: Biomass/biofuel burning also contributes to $Cl^-$ emissions in winter?
**Answer:** Yes, biomass and biofuel burning could also contribute to $Cl^-$ emissions in winter
(Christian et al., 2010; Li et al., 2014). However, the emission of biofuel burning is relatively
stable during the whole year and the average mass $Cl^-/K^+$ ratio of 7.1 (except for firework event
during the Spring Festival) in winter was about a factor of 2 greater than the value of 3.8 in
autumn when biomass (straw) burning was prevailing in the region. Therefore, the obvious
elevation of $Cl^-$ proportion in WSIs as well as the extremely high $Cl^-$ concentrations in winter
should be ascribed to the additional source of residential coal combustion.
**Comment 5:** Line 19: Mineral dust, including $Ca^{2+}$, was transported from farmland to urban
region? Construction activities also contributed to high values of $Ca^{2+}$ in urban region.
**Answer:** Yes, construction activities are an important source for atmospheric $Ca^{2+}$ in urban region.
However, there are few construction activities in the rural area, which couldn't explain the
extremely high concentrations of $Ca^{2+}$ over there during the autumn (Fig. R2). The extremely high
concentrations of $Ca^{2+}$ in Beijing occurred during the period of 6-25 October when the air parcels
were mainly from the southwest/south regions (Fig. R4) where the vast areas of agricultural field were under intensive maize harvest and soil ploughing. Although the concentrations of $Ca^{2+}$ in the rural area were still kept high levels during the period of 2-14 November (Fig. R2), the relatively low concentrations of $Ca^{2+}$ in Beijing were observed during the period when the air parcels were mainly from the northwest region (Fig. R4) where agricultural activities are relatively sparse. Considering the relatively stable contribution of construction activities to mineral dust during each season (Zhu et al., 2005), the coincident elevation of $Ca^{2+}$ in both the rural and urban areas and the evident increase of $Ca^{2+}$ proportion in WSIs of Beijing during the period of 6-25 October (Fig. R2 and Fig. R3) revealed the influence of the maize harvest and soil ploughing in the rural area on atmospheric $Ca^{2+}$ in Beijing.

**Comment 6:** Line 27, Note that industrial emissions from south regions in NCP are also massive.

**Answer:** There are massive industrial emissions from south regions in NCP. However, the emission of industries is relatively stable during the whole year (Gao et al., 2014), which cannot explain the distinct variations of the molar proportions of atmospheric WSIs in Beijing before, during and after the periods of heating in winter, maize fertilization in summer, and maize harvest and soil ploughing in autumn.

**Comment 7:** Line 36: $PM_{2.5}$ is not defined due to haze. The terminology should be clarified.

**Answer:** The mistake has been corrected in our revised manuscript: "The severe haze pollution is mainly ascribed to elevation of fine particulate matter with dynamic diameter less than 2.5μm ($PM_{2.5}$)".

**Comment 8:** Line 47-48: The authors should specify what traffic emissions included here, particles, gas, or both? Is it true that 4% of $PM_{2.5}$ was attributed to vehicle exhaust from Huang et al., 2014? This may be a wrong citation.

**Answer:** According to your valuable comments, we specify the traffic emissions. The traffic emissions reported in these references only included particles. Sorry, the reference should be Zhang et al., 2013. The mistakes have been corrected in our revised manuscript.

**Comment 9:** Line 55: How does this work advance our knowledge?

**Answer:** According to your valuable comments, the seasonal variation characteristics of WSIs in a rural site (Baoding, Hebei Province) have been added in our revised manuscript to advance our knowledge about the emissions from farmers' activities. Farmers' activities were found to make evident contribution to atmospheric WSIs in Beijing, based on the investigations about the seasonal variation characteristics of WSIs in both the rural and urban areas, and the distinct variations of the molar proportions of atmospheric WSIs in Beijing before, during and after the periods of heating in winter, maize fertilization in summer, and maize harvest and soil ploughing in autumn.

**Comment 10:** Line 65: totally?

**Answer:** "Totally" has been replaced with "mostly".

**Comment 11:** Line 73: What is the size of the particle on the crop leaves? More information is needed to show how long it can be transported. Also, wind speed is a key factor here.

**Answer:** The size of the particle on the crop leaves was not measured in this study. The previous studies confirmed that various plants can absorb atmospheric $PM_{2.5}$ and $PM_{10}$ (Bealey et al., 2007; Ji et al., 2013). There are about 300,000 $km^2$ agricultural fields where the harvest of wheat or maize mainly concentrates about two weeks in the NCP, and hence the emissions of mineral dust are suspected to be massive during the harvest through the harvest scene (Fig. R5). Although we don't know how long the particle from the harvest can be transported, the remarkable elevation of $Ca^{2+}$ proportion in Beijing during the period of the maize harvest and soil ploughing provided convincing evidences that the agricultural activities indeed influenced on atmospheric $Ca^{2+}$ in Beijing. Both wind speed and wind direction are indeed key factors for the transportation, while back trajectory is widely used for recognizing the transportation of pollutants. The extremely high concentrations of $Ca^{2+}$ in Beijing occurred during the period of 6-25 October when the air parcels were mainly from the southwest/south regions (Fig. R4) where the vast areas of agricultural field were under intensive maize harvest and soil ploughing. Although the concentrations of $Ca^{2+}$ in the rural area were still kept high levels during the period of 2-14 November (Fig. R2), the relatively low concentrations of $Ca^{2+}$ in Beijing were observed during the period when the air parcels were mainly from the northwest region (Fig. R4) where agricultural activities are relatively sparse. According to your valuable comments, we will perform the study about the size of the particle on the crop leaves in the near future.

**Comment 12:** Line 80: The authors should provide more solid evidence to show farmers' influences on an urban site in BJ?
Line 229-233: Again, more direct evidences are needed, if the authors wish to link the $Ca^{2+}$ in urban site to farmland.
Line 213-215: Detail explanation was needed here, how can the authors identify that coal combustion by farmers in winter might make great contribution to atmospheric $Cl^-$ other than coal combustion from urban area?

**Answer:** Solid evidence has been added in our revised manuscript (See the answers for comments 1, 3, 4 and 5).

**Comment 13:** Line 84: Can the contribution be quantified in this study?

**Answer:** It is difficult to quantify the contribution in this study because of the complex sources of atmospheric WSIs as well as the impact of meteorological factors. We are conducting the emission factors of various pollutants from typical farmers' activities such as residential coal combustion, the $NH_3$ emissions of agricultural field and so on, which will be helpful to quantify the contribution in the near future.

**Comment 14:** Line 91: Is this kind of filter suitable for the sampling at the site with high loading of $PM_{2.5}$?

**Answer:** The PTFE filter is widely used for PM sampling in previous studies (Chow et al., 1996; Walker et al., 2006; Pathak et al., 2009; Chen et al., 2015; Park et al., 2015). The significant correlation between WSIs sampled by the filters and $PM_{2.5}$ measured by the TEOM monitor (Fig. R6a), and the near equilibrium between cations and anions in the four seasons (Fig. R6b) indicated that this kind of filter is suitable for the sampling at the site with high loading of $PM_{2.5}$.

**Comment 15:** Line 93: Why started at 3 pm, background information is needed.
Line 99: How blank filters are sampled? It is better to show the blank values.

**Answer:** To conveniently replace the filter sample in each day, we select 3 p.m. as our starting time. Blank filters were brought to the field and were installed in the samplers which no air was pumped. After sampling, all the filters samples including blank filters were put in dedicated filter storage containers (90mm, Millipore) and preserved in a refrigerator till ion analysis. All the ion concentrations were corrected for blanks. The average blank values were about 0.03mg $L^{-1}$ for $Na^+$, $Ca^{2+}$, $F^-$, $NO_3^-$ and $SO_4^{2-}$, 0.02mg $L^{-1}$ for $NH_4^+$ and $Cl^-$, 0.01mg $L^{-1}$ for $Mg^{2+}$, $K^+$ and $HCOO^-$. According to your valuable comments, the blank values have been shown detailedly in our manuscript.

**Comment 16:** Line 114: How far is it from the observation site? Are the meteorological data and air pollutants similar at these two different sites?

**Answer:** There are about 20m between the observation station and our sampling site at almost the same height of 25m.

**Comment 17:** Line 116: Why 72h and 500m above sampling position were selected?

**Answer:** Due to the regional meteorological conditions with about 4-7 days periodic cycle (Guo et al., 2014), 72h is usually selected as the least elapsed time for recognizing regional transportation. Considering the surrounding terrain of Beijing and the height of planet boundary layer, air parcel with the height of 500m is recommended by NOAA for tracing their sources. In addition, the parameters have also been employed by previous studies (Li et al., 2012; Wang et al., 2015; Yang et al., 2016).

**Comment 18:** Line 128: Are there new findings by using this filter sampling method, compared with method described in section 2.1?
Line 135: How the mass of $PM_{2.5}$ filter was determined?
Line 140: Base on the comparison between filter sampling method and the TEOM 1405 Monitor, the authors can give out the underestimated percentage of concentrations of $PM_{2.5}$ and WSIs due to the volatile even semi-volatile component.
Line 154: This may be a good point to argue, but more details are needed.
Figure 1: The concentrations of WSIs are so high that close to $PM_{2.5}$. In general, organics are also as equally important as WSIs in Beijing, especially during days with lower PM$_{2.5}$. Mass closure
studies are needed to check the data quality.

**Answer:** As mentioned above, the comparison between the WSIs and PM$_{2.5}$ measured by the
TEOM monitor is far from the topic of the manuscript, and hence this part has been delated in our
revised manuscript.

**Comment 19:** Line 192: Why nitrate was faster than sulfate under higher pollution levels.
Line 195: Please show the pattern in different seasons.

**Answer:** The faster increase of nitrate proportion than that of sulfate proportion from clean days
to serious pollution days mainly occurred in spring and autumn when the concentration levels of
Ca$^{2+}$ were relatively high. To recognize the influence of Ca$^{2+}$ concentrations on the formation of
nitrate and sulfate, the formation rates of nitrate and sulfate were analyzed under typical cases of
haze formation in the four seasons (Fig. R7). It is evident that the faster formation rates of nitrate
than those of sulfate only occurred under the relatively high levels of Ca$^{2+}$ in spring and autumn,
indicating that the mineral dust could preferentially promote nitrate formation.

**Comment 20:** Line 217: Why the ratio Cl$^-$ to NO$_x$ was selected? They are different in phases in
the atmosphere.

**Answer:** NO$_x$ in Beijing is dominated by vehicles and relatively stable during the whole year.
Although Cl$^-$ and NO$_x$ are different in phases in the atmosphere, the Cl$^-$/NO$_x$ ratio value can
counteract the influence of meteorological factors and reveal the additional sources for
atmospheric Cl$^-$ in the four seasons. Considering this situation, the Cl$^-$/NO$_x$ ratio has been delated
in our revised manuscript.

**Comment 21:** Line 248-249: This is an important point and the evidence is critically needed.

**Answer:** NH$_3$ emissions generated from a prevailing residential coal stove fueled with raw
bituminous coal were investigated under alternation cycles of flaming and smoldering combustion
in our preliminary studies. The NH$_3$ emission factor for the residential coal stove was recorded as
0.62-1.10g/kg coal, which was in line with Li et al., 2016. These results indicated that residential
coal combustion may be a significant NH$_3$ emission source in the cold winter, and hence leading
to the elevation of atmospheric NH$_4^+$ in Beijing.

**Response to Reviewer #1**
**Comment 1:** So many studies have already been carried out for the chemical compositions from
PM$_{2.5}$ in Beijing. Because of the lack of other related aerosol measurements, this paper basically
focuses on the simple display of the ion concentrations. Most of the discussion are based on
speculation, and no new ideas and no interesting points are found in this paper. On the whole, this
paper is not suitable for publication in the ACP.

**Answer:** According to your comments, the seasonal variation characteristics of WSIs in a rural site (Baoding, Hebei Province) have been added in our revised manuscript to advance our knowledge about the emissions from farmers' activities. Farmers' activities were found to make evident contribution to atmospheric WSIs in Beijing, based on the investigations about the seasonal variation characteristics of WSIs in both the rural and urban areas, and the distinct variations of the molar proportions of atmospheric WSIs in Beijing before, during and after the periods of heating in winter, maize fertilization in summer, and maize harvest and soil ploughing in autumn.

Although we didn't conduct other related aerosol measurements, the variation characteristics of the composition of the water-soluble ions in $PM_{2.5}$ were found to well reflect their possible sources, and the following important conclusions were derived from the measurements: 1. Because the atmospheric $Cl^-$ sources from sea-salt, industries, power plants and biofuels are relatively stable during the whole year and the average mass $Cl^-/K^+$ ratio of 7.1 (except for firework event during the Spring Festival) in winter was about a factor of 2 greater than the value of 3.8 in autumn when straw burning was prevailing in the region, the obvious elevation of $Cl^-$ proportion (Fig. R3) as well as $Cl^-$ concentrations (Fig. R2) in winter should be ascribed to the additional source of residential coal combustion. Besides $Cl^-$, the serious emissions of various pollutants from residential coal combustion (Zhang and Tao, 2008; Zhang et al., 2008; Li et al., 2016) must make evident contribution to deteriorate the air quality in Beijing during the wintertime. 2. Compared with the periods before and after maize fertilization, the proportion of $Cl^-$ during maize fertilization in summer increased about 3%-4%, confirming the influence of maize fertilization on atmospheric $Cl^-$ in Beijing. Because fertilization is an important source for atmospheric $NH_3$, the elevation of $Cl^-$ (as a tracer for fertilization) revealed that fertilization in the rural areas around Beijing could also make obvious contribution to atmospheric $NH_4^+$ in Beijing. 3. The remarkable elevation of $Ca^{2+}$ proportion in Beijing during the period of the maize harvest and soil ploughing provided convincing evidences that the agricultural activities indeed influenced on atmospheric $Ca^{2+}$ in Beijing. With the elevation of $Ca^{2+}$ in spring and autumn, the evidently faster increasing rates of $NO_3^-$ than $SO_4^{2-}$ implied that the atmospheric heterogeneous reaction of $NO_2$ on the mineral dust was an important source for $NO_3^-$. To our best knowledge, there are still no reports about the above conclusions which will be helpful for future control measures in reducing pollutant emissions from rural areas in the North China Plain. Additionally, the heterogeneous reaction of $NO_2$ on mineral dusts has been found to make contribution to nitrate formation under laboratory simulations, but the role of the reaction for nitrate formation has not been recognized in field measurements before this study. Because field measurement is one of the main subject areas of the ACP and there are original findings in the paper, we think the paper is suitable for publication in the ACP.

**Comment 2:** In addition, there are also some problems and mistakes in this paper. After major revision, this paper might be suitable for publication in some local journals. It is strongly recommended that this paper be send to a language editing service. There are too many Chinese English in this paper. For example, the use of the word "farmer" is inaccurate, even ridiculous, just as "with high density of famers", "farmers' activities", "heating by farmers". At present, most of the people living in the rural area are not engaged in agricultural activities. And farmers have also not engaged in agricultural activities in most of the time. You should use the "rural area" and "agricultural activities" to describe the exact meaning.

**Answer:** Thank you for your valuable comments. We have revised our manuscript and corrected some mistakes in this paper. However, to our best knowledge, "farmer" is a commonly used word to represent for people who are living in rural areas (Xhoxhi et al., 2014; Pattey and Qiu, 2012; Rio et al., 2011; Mahmud, 2009) and the "rural area" was also used for representing the countryside in the text of the paper. In addition, the "farmers' activities" in this paper included both "agricultural activities" and farmers' living activities (cooking and heating via coal combustion, etc.), and hence "farmers' activities" is more exact than "agricultural activities" for describing our meanings.

**Comment 3:** Line 70, "Because crop leaves absorbed large quantities of atmospheric particles during crop growing season, the abrupt release of the particles by smashing crop straw for returning in the vast area of the NCP must also make striking contribution to atmospheric particles in the region during the seasonal harvest seasons." This statement is basically impossible to be true. There is no evidence that the crop could absorb $PM_{2.5}$. And the smashing process of crop straw could not be an important source of $PM_{2.5}$. Just a small amount coarse PM might be emitted.

**Answer:** The previous studies have confirmed that various plants can absorb atmospheric $PM_{2.5}$ and $PM_{10}$ (Bealey et al., 2007; Ji et al., 2013). There are about 300,000 $km^2$ agricultural fields where the harvest of wheat or maize mainly concentrates about two weeks in the NCP, and hence the emissions of mineral dust are suspected to be massive during the harvest through the harvest scene (Fig. R5). In addition, the remarkable elevation of $Ca^{2+}$ proportion in Beijing during the period of the maize harvest and soil ploughing as well as the back trajectory cluster analysis (See the answers of comment 5 of Reviewer #2) provided convincing evidences that the agricultural activities indeed influenced on atmospheric $Ca^{2+}$ in Beijing.

**Comment 4:** Line 74, what's the meaning of "pollutant emissions from the chimney of the farmers' coal stoves"? There is not a thing called "farmers' coal stoves" in this world. I think "pollutants from coal combustion for heating" is more accurate. The author is not familiar with the countryside.

**Answer:** Because "pollutants from coal combustion for heating" includes various sources from industrial boilers, central-heating boilers as well as residential coal stoves, this word might be difficult to describe the meaning of the sentence accurately. According to your suggestion, the revised manuscript has replaced "the farmers' coal stoves" with "the residential coal stoves". It should be mentioned that the corresponding author of this paper was born and grew up in a village of the North China Plain and frequently visits the village every year. In addition, our group has been engaged in field measurements of $N_2O$ emissions for about ten years. Therefore, we are familiar with the rural areas very well.

**Comment 5:** Line 94, "dedicated filter storage containers"? I think it should be a desiccator.

**Answer:** The dedicated filter storage container is not a desiccator but a kind of dedicated box for storing the filters. The objective of this paper is to investigate the water-soluble ions in $PM_{2.5}$ not to measure the mass concentrations of $PM_{2.5}$, and hence desiccators were not used as containers for the filters. The dedicated filter storage containers are commercial products which have been widely used for storing the filters by investigators.

**Comment 6:** As mentioned in this paper, the TEOM 1405 is not suitable for accurate $PM_{2.5}$ mass concentration measurement owing to the volatilization of unstable components. Why didn't the authors weigh the PTFE filters before and after the sampling for mass concentration analysis? This is the biggest problem in this paper. The proportions of different ions in $PM_{2.5}$ could not be obtained.

**Answer:** In this paper, the variation characteristics of the water-soluble ions could provide the important information about the evident contribution of farmers' activities. The comparison between the WSIs and $PM_{2.5}$ measured by the TEOM monitor is far from the topic of the manuscript, and hence this part has been delated in our revised manuscript. According to your suggestions, we will perform mass closure studies in the near future.

**Response to SC#1**

**Comment 1:** The manuscript presents results from ion chromatography analysis of samples of $PM_{2.5}$ collected in Beijing through the year of 2014 aimed at deriving the variation characteristics of water-soluble ions (WSIs) in the $PM_{2.5}$. Since only a small part of studies focused on the variation characteristics of WSIs in the four seasons by now, I think the intentions of the authors are very good and substantial data about the WSIs in the $PM_{2.5}$ are provided which can make an incremental gain in the knowledge of the haze occurred in Beijing. The science is sound and the results are meaningful. In addition, the authors are very familiar with the North China plain (NCP) and the agriculture activities and living activities of farmers in NCP. There are interesting findings that the emissions from farmers' activities in the NCP was one possible emission sources and the influence of fertilization events and crop straw have influence on the regional air quality during the harvest seasons periods which have been neglected by most previous studies. Maybe more attention would be paid to the agriculture activities in NCP and that some field observations would be carried out in rural area after this paper. The detailed data of the daily variations of WSIs in this paper showed an obvious seasonal variation characteristic, which may helpful for further exploring how meteorological factor affect the accumulation and dispersion of atmospheric pollutants.

**Answer:** Thank you for your approval and your valuable comments. Just as you know, the corresponding author of this paper was born and grew up in a village of the North China Plain and frequently visits the village every year. In addition, our group has been engaged in field measurements of $N_2O$ emissions for about ten years. Therefore, we are familiar with the rural areas very well. In recent years, our observation found that the frequent haze formation periods closely relate with the periodically strong emissions of pollutants from the rural area in the NCP, which mainly occurred in summer season of June-July during wheat harvesting period, autumn season of September-October during maize harvesting period and winter season during the heating period by residential coal combustion. Considering that the emissions from the rural area in the NCP are almost neglected, we presented the new ideas and the interesting points by tracing the sources of atmospheric WSIs in $PM_{2.5}$. The further exploring has been performed during the whole year 2015 and the contribution of periodic emissions from farmers' activities would be quantified in the near future.

**Comment 2:** And it was found that the atmospheric concentrations of $SO_2$ and $NO_2$ in autumn are much smaller than that in winter and spring, whereas the mean concentration of WSIs in autumn was almost the same as that in winter and nearly twice as that in spring. This result indicates that unknown mechanisms of atmospheric heterogeneous reactions and transformation of atmospheric pollutants from gas phase to particulate phase should be investigated. Moreover, it was an interesting observation that the increasing rates of $SO_4^{2-}$ during some serious pollution events were much slower than those of $NO_3^-$, especially with the elevation of $Ca^{2+}$. The heterogeneous reactions of $SO_2$ and $NO_2$ with mineral dust may be an important pathway for the formation of sulfate and nitrate in the urban cities of East Asia because of the frequent occurrence of dust storms. Most previous studies focused on the heterogeneous uptake of $SO_2$ or $NO_2$ on mineral aerosol surfaces without considering coexistent gases in atmospheric condition. Only a few studies reported that $SO_2$ and $NO_2$ likely exert synergistic effect on the surface of mineral dust. To my knowledge there is still a lack of knowledge to explain why the increase of nitrate proportion with increasing pollution levels much faster than the increase of sulfate. I'm interested in the new ideas and inspiring points in this paper.

**Answer:** We entirely agree with your comments. The processes and evolution of haze pollution are characterized by the formation of substantial amounts of sulfate and nitrate (Sun et al., 2006; Zhao et al., 2013). The large amount of sulfate and nitrate were considered to be more likely generated via heterogeneous chemistry than gas-phase and aqueous-phase chemistry during haze days in China (Zhao et al., 2013; Wang et al., 2013). Modeling studies and laboratory simulations have researched on the role of heterogeneous reactions in sulfate and nitrate formation on the surface of mineral particles (Zheng et al., 2015), but the role of the reaction for nitrate formation has not been recognized in field measurements before this study.
The faster increase of nitrate proportion than that of sulfate proportion from clean days to serious pollution days mainly occurred in spring and autumn when the concentration levels of $Ca^{2+}$ were relatively high. To recognize the influence of $Ca^{2+}$ concentrations on the formation of nitrate and sulfate, the formation rates of nitrate and sulfate were analyzed under typical cases of haze formation in the four seasons (Fig. R7). It is evident that the faster formation rates of nitrate than those of sulfate only occurred under the relatively high levels of $Ca^{2+}$ in spring and autumn, indicating that the mineral dust could preferentially promote nitrate formation. However, the reason might be further analyzed by laboratory simulation in the near future. Thank you.

[Figure]

**Fig. R1** Sampling sites (the urban site in Beijing city and the rural site in Baoding, Hebei Province) in the NCP.

[Figure]

**Fig. R2** Seasonal variations of the several typical WSIs in the year of 2014. (The mass concentrations of Cl⁻, K⁺,

Ca²⁺, NH₄⁺, SO₄²⁻ and NO₃⁻ were presented at RCEES and DBT. The green square showed the firework event during the period of the Spring Festival. The gray square represented farmers' activities, including residential coal combustion for heating (a), top dressing for wheat (b), wheat harvest and basal fertilization for maize (c), top dressing for maize (d), maize harvest and soil ploughing (e) and straw burning (f).)

[Figure]

**Fig. R3** Molar proportions of atmospheric WSIs in Beijing before, during and after the periods of heating in winter, maize fertilization in summer, and maize harvest and soil ploughing in autumn.

[Figure]

**Fig. R4** The back trajectory cluster analysis and the corresponding overall ion mass concentration in four seasons.

[Figure]

**Fig. R5** The harvest scene during the wheat harvest of 2014 in the rural area (close to our rural site) in Baoding,

                            Hebei Province.

[Figure]

[Figure]

**Fig. R6** The correlation between WSIs sampled by the filters and PM$_{2.5}$ measured by the TEOM monitor (Fig.

R6a, 1-24 January, 2015), and the ratios of cations to anions in the four seasons of 2014 (Fig. R6b).

[Figure]

**Fig. R7** Case studies about the increasing rates of $NO_3^-$ and $SO_4^{2-}$ with the elevation of $Ca^{2+}$ during serious pollution events in the four seasons.

A list of all relevant changes made in the manuscript

Based on the valuable comments and suggestions of the three reviewers, the followings are a list of all relevant changes made in the manuscript.

1. The data of WSIs in $PM_{2.5}$ at the rural site during the year 2014 has been added in our revised manuscript.

2. Solid evidences about the impacts of farmers' activities ($Cl^-$ from coal combustion by farmers, fertilization of $NH_4Cl$ as well as $Ca^{2+}$ from maize harvest and soil ploughing and so on) on regional air quality have been added in our revised manuscript.

3. The molar composition of WSI under different pollution levels and the comparison between WSIs and $PM_{2.5}$ have been delated in our revised manuscript due to being away from the topic of the manuscript.

4. The specification about the data quality assurance has been added in our revised manuscript.

5. Most of figures have been amended for supporting the results and discussion in the manuscript.

6. Some logical and grammatical mistakes have been corrected in our revised manuscript.

7. Several references have been inserted to confirm our points in our revised manuscript.

[revised manuscript text omitted]

---

## Editor Decision (ED1)

In my opinion, the revised version of the manuscript has improved quite a bit. The authors have done a good job addressing the multiple concerns raised by the reviewers. I have a few remaining comments, mostly of technical nature:

**Throughout the text**: insert spaces between values and their units, for example, 170km -> 170 km

**Figures**: some of the figures in the PDF appear to have low resolution and are difficult to read without zooming into the figures. In particular, some of the axis labels, axis tick labels, and legends appear blurry. I would work on optimizing the resolution as well as size of the text used for annotating the figures. Figures 5 and 10 are especially difficult to read because of its large information content.

Line 84: the "intensity" of the emission is ambiguous; it is better to compare actual emissions factors in g pollutants per kg of burned coal. I am sure those have been measured.

Line 107: what is meant by "artificial intelligence" here? I looked up the info about the Laoying 2034 sampler, it appears to be a normal sampling instrument.

Line 132: please confirm that the station is only 20 m away. Do you perhaps mean 20 km?

Line 141: I think you should define what you mean by the "total cation concentration" in this sentence. Do you account for the different charge states of the ions [total positive] = $[Na^+]$ + $[NH_4^+]$ + $2*[Mg^{2+}]$ + $2*[Ca^{2+}]$ + $[K^+]$? Also, since your positive and negative ions appear to be balanced, does it mean that your PM2.5 is always neutralized (not acidic, with very low $[H^+]$)? If so this would be worth discussing because particle acidity is an important parameter in controlling SOA growth on particles.

Line 156: The proposed explanation for the lower than expected PM2.5 mass measured with TEOM needs more support. Instead of citing the Finlayson-Pitts book, please provide references proving that $NH_4NO_3$ and other volatiles are indeed depleted from PM2.5 measured with TEOM. I find it difficult to believe that 50°C would depletes things other than water from particles to a measurable extent. I suspect that other readers will also have doubts about that. So more references here would definitively help.

Line 158: is the 20% value based on your measurement done in this work or on measurements done by Yang et al. (2015)?

Line 163: the variation is **not** periodic, please see the suggested correction in the table below.

Lines 206 and 212: you are using molar ratios in some cases and mass ratios in others. It would help to be more uniform to avoid confusion.

Line 222: specify the amount burned in kg (or another appropriate SI unit) per year; use an appropriate unit modifier to get rid of the trailing zeros in the number. Example: 42 Tg/year

Line 232: since you are attributing high Cl$^-$ to the use of NH$_4$Cl fertilizer, it would be useful to discuss a correlation between the chloride and ammonium ions in this period compared to other periods.

The use of English in this paper will need to be improved before the paper can appear in its final form in ACP. The table below lists some of the mistakes but it is not a comprehensive list. Given the high number of mistakes I am going to have to request a proof-reading service from the journal.

| Line | Action | Text | New text |
|---|---|---|---|
| 11 | Replace | including Beijing | ,which includes Beijing, |
| 12 | Replace | status | problem |
| 14 | Replace | as well as a rural | and in a rural |
| 15 | Delete | characteristics | |
| 15 | Replace | the PM2.5 | PM2.5 |
| 16 | Replace | for recognizing | to determine |
| 17 | Replace | made evident | made a significant |
| 19 | Replace | made evident contribution | contributed |
| 21 | Replace | were reasonably | could be |
| 24 | Delete | rationally | |
| 27 | Replace | evidences | evidence |
| 28 | Replace | made evident contribution | contributed |
| 34 | Replace | regions with | regions, which have a |
| 47 | Replace | status | problem |
| 59 | Replace | from the | from the emissions from |
| 60 | Replace | on both | in both |
| 62 | Replace | almost | often |
| 63 | Replace | agricultural | of agricultural |
| 64 | Delete | very | |
| 65 | Replace | focus on | occur in |
| 76 | Replace | prevailing | prevalent |
| 80 | Replace | striking | a |
| 81 | Replace | chimney | chimneys |
| 82 | Rephrase | I do not understand what you mean by "imagined by the strong smog" | |
| 82 | Replace | small | a small |
| 88 | Replace | daily collected | collected daily |
| 91 | Replace | evidences | evidence |
| 95 | Replace | A sampling … was chosen on | The sampling … was on |
| 117 | Replace | water | a |
| 147 | Replace | could well reveal the pollution status | could be used as an indicator of the pollution level |
| 149 | Delete | much | |
| 150 | Replace | Therefore, the … was suspected to be largely | It is possible that the … was underestimated |

| | | underestimated | |
|---|---|---|---|
| 157 | Replace | accounts | account |
| 161 | Replace | variations | variation |
| 161 | Replace | statistic | average |
| 163 | Replace | daily variations of the WSIs at RCEES exhibited significantly periodic fluctuation | the concentrations of the WSIs varied greatly on timescale of days |
| 165 | Replace | the most frequently high | the highest |
| 167 | Replace | pollutants | pollutants' |
| 176 | Replace | fast thermal decomposition | reduced gas-to-particle partitioning |
| 179 | Replace | remarkable | large |
| 180 | Replace | would override the relatively low atmospheric photo-oxidants for their oxidation rates | would result in large sulfate and nitrate formation rates despite the lower concentrations of oxidizing species |
| 181 | Replace | resulted | result |
| 188 | Replace | seasonal variation characteristics | concentrations |
| 189 | Replace | comparatively illustrated | compared |
| 196 and 197 | Replace | parcel | parcels |
| 199 | Replace | made evident contribution | contributed |
| 201 | Replace | make evident contribution | affect |
| 203 | Replace | Without considering | With the exception of |
| 203 | Replace | concentration | concentrations |
| 218 | Replace the http link with a reference to one | | |
| 227 | Replace | to be noted | to note |
| 229 | Replace | make contribution | contribute |
| 238 | Replace | , 2016 | (2016) |
| 246 | Replace | storm | storms |
| 249 | Replace | make contribution | contribute |
| 255 | Replace | cultivation manner | method |
| 266 | Replace | were still kept high levels | remained at high levels |
| 262-269 | Split this very long sentence in 2 or 3 sentences | | |
| 276, 278 | Replace | frequently | frequent |
| 276 | Replace | slow thermal decomposition | increased gas-to-particle partitioning at lower temperatures |
| 283 | Replace | magnitude | order of magnitude |
| 286 | Replace | water of particulate matters | aerosol water |
| 305 | Delete | well | |

| 316 | Replace | to be noted | to note |
|---|---|---|---|
| 337 | Replace | conspicuous | large |
| 341 | Replace | strongly periodic activities of farmers | farmers' activities |
| 342 | Replace | make evident contribution | contribute |
| 344 | Replace | aroused great attention | paid greater attention |
| 571 | Replace | Daily variations | Variation |
| 571 | Replace | The smooth | the smooth |
| 576 | Replace | Seasonal variations of the several typical WSIs | Concentrations of selected WSIs |
| 578 | Replace | grey square represented | areas shaded in yellow represent |
| 593 | Replace | proportions | fractions |
| 598 | Replace | concentration | concentrations |

---

## Author Response (AR2)

A point-by-point response to the reviews

Thank you for your valuable comments. The followings are our responses to your comments.

**Response to Reviewer #3**

**Comment 1:** The article aims to analyze the seasonal variation characteristics of the typical WSIs (WSIs) in the $PM_{2.5}$ collected at an urban site in Beijing as well as a rural site in Baoding. The results from ion chromatography analysis of daily samples through the year of 2014 were presented in this study, the field and lab experiment is laborious. The data collected at the two sites were analyzed well and new ideas were given. The authors proposed the possible contribution of the periodic emissions from farmers' activities in the North China Plan (NCP) to the WSIs in Beijing, and the influence of fertilization events and crop straw had important influence on the regional air quality during the harvest seasons periods. Also, the maize harvest and soil ploughing in autumn in NCP

were suspected to make contribution to the atmospheric $Ca^{2+}$ in Beijing, which has not been explained well by previous studies. Since the influence of fertilization events and crop straw on the air quality have been neglected by most previous studies and only a small part of studies focused on the variation characteristics of WSIs in the four seasons, the detailed data of the daily variations of

WSIs in this article are helpful for further identification of the pollution sources and how meteorological factor affect the accumulation of atmospheric pollutants.

**Answer:** Thank you for your extremely positive evaluation of our work. The followings are our responses to your comments.

**Comment 2:** The data in this article are collected at analyzed from two sites in Beijing and Baoding, and the major findings in this study is the possible contribution of the periodic emissions from farmers' activities in the NCP, such as crop harvest, crop straw burning, and coal combustion, to the WSIs in Beijing, while the title of this paper is "The variation characteristics and possible sources of atmospheric water-soluble ions in Beijing", the title is confusing and I wonder whether it is accurate. Maybe the inspiring points about the possible contribution of the periodic emissions from farmers' activities in the NCP could be highlighted.

**Answer:** Thank you for your valuable comments. The title has been replaced with "The possible contribution of the periodic emissions from farmers' activities in the North China Plain to atmospheric water-soluble ions in Beijing".

**Comment 3:** There are some confusing expressions in this article.

Line 657: residential coal stoves,

Line 666: after harvest of the winter wheat in June-July?

Line 673-674: the serious pollutant emissions from the chimney of the farmers' coal stoves can be easily imagined by the strong smog,

Line 842: intensive farmers' activities,

Line 915: Tham et al., 2016

**Answer:** These confusing expressions as well as other mistakes have been corrected in our revised manuscript:
"Line 657: farmers' coal stoves;
Line 666: after the wheat harvest (in June-July);
Line 673-674: heavy smoke from the chimneys of the farmers' coal stoves can be seen everywhere
in rural areas of the NCP due to heating supply;
Line 842: during the period of farmers' activities;
Line 915: Tham et al. (2016)"
**Response to Co-Editor**
**Comment 1:** In my opinion, the revised version of the manuscript has improved quite a bit. The
authors have done a good job addressing the multiple concerns raised by the reviewers. I have a few
remaining comments, mostly of technical nature:
**Answer:** Thank you for your appreciation. The followings are our responses to your comments.
**Comment 2: Throughout the text:** insert spaces between values and their units, for example,
170km $\rightarrow$ 170 km
**Answer:** Thank you for your valuable guidance. Spaces have been inserted between values and
their units throughout the text.
**Comment 3: Figures:** some of the figures in the PDF appear to have low resolution and are difficult
to read without zooming into the figures. In particular, some of the axis labels, axis tick labels, and
legends appear blurry. I would work on optimizing the resolution as well as size of the text used for
annotating the figures. Figures 5 and 10 are especially difficult to read because of its large
information content.
**Answer:** Thank you for your comments. All of the figures have been optimized: The axis labels,
axis tick labels and legends have been enlarged and in bold for reading easily without zooming into
the figures. Some unrelated information (such as MGL, the values of the increasing rates of $NO_3^-$
and $SO_4^{2-}$) in figures 5 and 10 has been delated and the necessary annotations have been added in
the caption (figure 5) in our revised manuscript.

| The number of the figure | Action | Treatment |
| --- | --- | --- |
| 1 | Optimize | Change the display order of figures |
| 2 | Optimize | Change the display order of figures |
| 3 | Optimize | Enlarge the sizes of the axis labels and legends |
| 4 | Optimize | Enlarge the sizes of the axis labels and legends |
| 5 | Revise and annotate | Delate the unrelated information, change the display order and add the annotations |
| 6 | Revise | Change the colors and increase the font sizes |
| 7 | Optimize | Change the arrows in bold |
| 8 | Optimize | Change the symbols in bold |
| 9 | Optimize | Enlarge the sizes of the axis labels and legends |

| 10 | Revise | Delate the unrelated information, change the colors and enlarge the sizes of the axis labels and legends |

**Comment 4:** Line 84: the "intensity" of the emission is ambiguous; it is better to compare actual emissions factors in g pollutants per kg of burned coal. I am sure those have been measured.

**Answer:** The emissions factors of typical pollutants have been compared and added in our revised manuscript: "…the emission factors of typical pollutants such as $PM_{2.5}$, organic carbon (OC) and polycyclic aromatic hydrocarbons (PAHs) from farmers' coal stoves (about 1054-12910 mg/kg for $PM_{2.5}$, 470-7820 mg/kg for OC and 58.5-229.1 mg/kg for PAHs) are usually about 1-3 orders of magnitude greater than those from coal power plants or industries boilers (about 16-100 mg/kg for $PM_{2.5}$, 0.3-17.1 mg/kg for OC and 0.8-12.8 µg/kg for PAHs) (Zhang et al., 2008; Xu et al., 2006; Geng et al., 2014; Chen et al., 2005; Revuelta et al., 1999; Yang et al., 2016)…"

**Comment 5:** Line 107: what is meant by "artificial intelligence" here? I looked up the info about the Laoying 2034 sampler, it appears to be a normal sampling instrument.

**Answer:** Thanks for your comments. "Artificial intelligence" has been delated in our revised manuscript.

**Comment 6:** Line 132: please confirm that the station is only 20 m away. Do you perhaps mean 20 km?

**Answer:** The Beijing urban ecosystem research station is indeed 20 m away from our sampling site of RCEES because the station is also located in our institute (RCEES) and the distance between them is the spacing (about 20m) between two laboratory buildings.

**Comment 7:** Line 141: I think you should define what you mean by the "total cation concentration" in this sentence. Do you consider the different charge states of the ions [total positive] = $[Na^+]$ + $[NH_4^+]$ + $2*[Mg^{2+}]$ +$2*[Ca^{2+}]$ + $[K^+]$? Also, since your positive and negative ions appear to be balanced, does it mean that your $PM_{2.5}$ is always neutralized (not acidic, with very low $[H^+]$)? If so this would be worth discussing because particle acidity is an important parameter in controlling SOA growth on particles.

**Answer:** Yes, the different charge states of the ions had been considered in our manuscript. Although the positive and negative ions appear to be balanced, it doesn't mean that the $PM_{2.5}$ is always neutralized because of the unknown concentrations of $CO_3^{2-}$ and $HCO_3^-$ (Hennigan et al., 2015). If we consider the unknown concentrations of $CO_3^{2-}$ and $HCO_3^-$, the $PM_{2.5}$ should be acidic (with $[H^+]$). In addition, several studies have estimated the concentrations of $H^+$ by using the formula below (with the exception of $Mg^{2+}$ and $Ca^{2+}$) (Zhang et al., 2007; Behera et al., 2015):

$$[H^+] = [Cl^-] + [NO_3^-] + 2 \times [SO_4^{2-}] - [NH_4^+] \tag{1}$$

or

$$[H^+] = [Cl^-] + [NO_3^-] + 2 \times [SO_4^{2-}] - [NH_4^+] - [Na^+] \tag{2}$$

It is evident that the $PM_{2.5}$ in this study will be acidic based on the two methods. Particle acidity is indeed an important parameter in controlling SOA growth on particles, which has been reported by previous studies (Zhang et al., 2004; Hu et al., 2014). According to your valuable comments, the "total cation concentration" and "total anion concentration" have been defined as "$[Na^+] + [NH_4^+] + 2 \times [Mg^{2+}] + 2 \times [Ca^{2+}] + [K^+]$" and "$[F^-] + [HCOO^-] + [Cl^-] + [NO_3^-] + 2 \times [SO_4^{2-}]$" in our revised manuscript, respectively.

**Comment 8:** Line 156: The proposed explanation for the lower than expected $PM_{2.5}$ mass measured with TEOM needs more support. Instead of citing the Finlayson-Pitts book, please provide references proving that $NH_4NO_3$ and other volatiles are indeed depleted from $PM_{2.5}$ measured with TEOM. I find it difficult to believe that 50 ℃ would deplete things other than water from particles to a measurable extent. I suspect that other readers will also have doubts about that. So more references here would definitively help.

**Answer:** Thank you for your recommendation. Another definitive reference (Charron et al., 2004) has been found to confirm our proposed explanation for the lower than expected $PM_{2.5}$ mass measured with TEOM. Charron et al. (2004) carried out a comparison of $PM_{10}$ and $PM_{2.5}$ mass measured simultaneously by TEOM and filter-based gravimetric methods. The results confirmed the expected large difference between the two methods and large differences between measurements are associated with high particulate ammonium nitrate concentrations. According to your comments, this reference has been added in our revised manuscript and uploaded in the system as the supplement for your check.

**Comment 9:** Line 158: is the 20% value based on your measurement done in this work or on measurements done by Yang et al. (2015)?

**Answer:** The 20% value is based on your measurement done in this work. This part has been made clear in our revised manuscript: "…whereas they were found to only account for about 20% in the filters of the TEOM 1405 Monitor in this study…"

**Comment 10:** Line 163: the variation is **not** periodic, please see the suggested correction in the table below.

**Answer:** The mistake has been corrected in our revise manuscript based on your valuable comments. Thank you very much.

**Comment 11:** Lines 206 and 212: you are using molar ratios in some cases and mass ratios in others. It would help to be more uniform to avoid confusion.

**Answer:** Thank you for your valuable comments. Molar ratios were employed in most cases in the paper, thus mass ratios (the mass $Cl^-/K^+$ ratios in Line 212 and the mass concentration ratios of $NO_3^-/SO_4^{2-}$ in Table 3) have been replaced with molar ratios in our revised manuscript.

**Comment 12:** Line 222: specify the amount burned in kg (or another appropriate SI unit) per year; use an appropriate unit modifier to get rid of the trailing zeros in the number. Example: 42 Tg/year

**Answer:** Thank you for your valuable guidance. 42,000,000 Tons (line 222) and 1,174,000 Tons
(line 232) have been replaced with 42 Tg/year and 1.174 Tg.

**Comment 13:** Line 232: since you are attributing high $Cl^-$ to the use of $NH_4Cl$ fertilizer, it would
be useful to discuss a correlation between the chloride and ammonium ions in this period compared
to other periods.

**Answer:** The correlation between $Cl^-$ and $NH_4^+$ might be not significant in this period compared to
other periods, because $NH_4^+$ was not only from the direct emission of $NH_4Cl$ fertilizer but also
dominated by the concentrations of atmospheric $NH_3$. The soil source (microbial activities) and the
animal wastes mainly contributed to atmospheric $NH_3$, especially during the summer with high
temperatures (Krupa, 2003). Therefore, it might be difficult to discuss the correlation between $Cl^-$
and $NH_4^+$ for recognizing the source of $NH_4Cl$ fertilizer.

**Comment 14:** The use of English in this paper will need to be improved before the paper can appear
in its final form in ACP. The table below lists some of the mistakes but it is not a comprehensive
list. Given the high number of mistakes I am going to have to request a proof-reading service from
the journal.

**Answer:** We think it is an excellent arrangement for requesting a proof-reading service from the
journal. According to your comments, both the revised manuscript (marked) and a clean one have
been uploaded in the system. Thank you very much.

**Comment 15:** (from the Table in the review)
Line 82: Rephrase "…imagined by the strong smog…"
Line 218: Replace the http link with a reference to one
Line 262-269: Spilt this very long sentence in 2-3 sentences

**Answer:** Thank you very much for your careful reviews. These sentences have been corrected in
our revised manuscript:
"Line 82: heavy smoke from the chimneys of the farmers' coal stoves can be seen everywhere in
rural areas of the NCP due to heating supply;
Line 218: the http link has been replace with the reference of Ma et al. (2015);
Line 262-269: The back trajectory cluster analysis also supported the above conclusion: (1) the
extremely high concentrations of $Ca^{2+}$ in Beijing occurred during the period of 6-25 October (Fig.
3 and Fig. 4) when the air parcels were mainly from the southwest/south regions (Fig. 5) where the
vast areas of agricultural field were being under intensive maize harvest and soil ploughing. (2)
Although the concentrations of $Ca^{2+}$ in the rural area remained at high levels during the period of 2-
14 November (Fig.3 and Fig. 4), the relatively low concentrations of $Ca^{2+}$ in Beijing were observed
during the period when the air parcels were mainly from the northwest region (Fig. 5) where
agricultural activities are relatively sparse."
In addition, other revisions such as "replace" and "delate" listed in the Table have been done in our revised manuscript. Thank you for all you've done for us.

instruments for many years at more than 350 sites in the UK, including the 50 urban background, kerbside, industrial and rural sites belonging to the Automatic Urban and Rural Network (AURN). The TEOM instrument has the advantage over conventional gravimetric methods of particle mass monitoring to provide data on an almost real-time basis and to be a cost and labour-effective method. Their introduction in the UK has led to significant improvements in knowledge of the sources and behaviour of particulate matter (QUARG, 1996; APEG, 1999). In comparison, 24-h measurements with filter-based gravimetric methods are costly and time consuming. The filter handling involves a large number of steps including pre-conditioning, weighing of blanks, filter installation and filter removal on the sampling site, post-conditioning and weighing of dust-loaded filters.

However, many studies comparing various filter-based $PM_{10}$ (or $PM_{2.5}$) samplers with TEOM instruments have shown that TEOMs report lower particle mass values than the collocated filter-based samplers (Allen et al., 1997; Ayers et al., 1999; Soutar et al., 1999; APEG, 1999; Salter and Parsons, 1999; Williams and Bruckmann, 2001; Cyrys et al., 2001). This is attributed to heating of the inlet of the TEOM system, usually to $50°C$, conducted initially in order to minimise interferences from the evaporation and condensation of water onto the filter and to provide a stable and reproducible measurement (Patashnick and Rupprecht, 1991). The particulate material lost in the inlet of the TEOM is thought to be mainly semi-volatile particulate matter and particle-bound water, both of which partition between condensed and vapour phases in the atmosphere. The major such components are expected to be ammonium nitrate and semi-volatile organic compounds.

It is important to highlight that filtration-based mass measurements are not artefact-free. They can also lead to significant volatilisation losses of the same semi-volatile components during collection, due to changes in temperature, relative humidity, composition of the aerosol or pressure drop across the filter, and after collection—due to filter handling, transport and storage (see Chow, 1995). Artefacts in the measurement of particle mass concentrations from gravimetric methods also arise from the adsorption of semi-volatile organic gases onto or from collected particulate matter and filter media (Turpin et al., 2000) and the neutralisation of acid or basic gases on either filter media, or collected particulate matter (Tsai and Huang, 1995). Artefacts may also be associated with particle-bound water that may constitute a significant part of particulate matter. The variability of the water content of particles and the hysteresis in the water adsorption–desorption pathways of some major atmospheric compounds (Seinfeld and Pandis, 1996) complicate the mass measurement. In recent works, Speer et al. (2003) show that water associated with both inorganic and hygroscopic organic components of $PM_{2.5}$ can contribute significantly to mass and Price et al. (2003) reveal that particle-bound water may be the major cause of difference between TEOM and reference gravimetric measurements.

The amounts of semi-volatile compounds and particle-bound water vary both temporally and spatially. As a consequence, the extent of the difference between TEOMs and filter-based gravimetric methods is not universal, and spatial and seasonal differences have been shown by many studies (Allen et al., 1997; APEG, 1999; Williams and Bruckmann, 2001) implying that the relationship between TEOMs and filter-based gravimetric methods should be examined for each site. This also implies that the relationship between TEOMs and gravimetric methods may not be proportional or linear and may be complex.

This paper describes a study of the relationship between $PM_{10}$ and $PM_{2.5}$ mass measurements by TEOM and Partisol (gravimetric) instruments at a rural site, and an investigation of the causal factors.

**2. Material and methods**

**2.1. Description of the site**

The monitoring station is within a self-contained, air-conditioned building located in the Harwell Research Science Centre in Oxfordshire. The nearest road is for access to buildings within the Science Park only. The manifold inlet is approximately 3 m above ground level. The surrounding area is generally open with agricultural fields. The nearest trees are at a distance of 200–300 m from the monitoring station. The station is a rural site that may be influenced by the London urban plume and possibly the busy A34 highway at a distance of 2 km to the east.

**2.2. Description of instruments and gravimetric measurements**

**2.2.1. Tapered element oscillating microbalance**

Two 1400A series TEOMs (Rupprecht and Patashnick Co., Inc.) have been providing measurements of $PM_{10}$ and $PM_{2.5}$ at Harwell on an almost real-time basis for a number of years. The particle mass is determined by continuous weighing of particles deposited onto a filter. The filter is attached to a vibrating hollow tapered glass tube. The frequency of mechanical oscillation of this tube is a function of its mass. Deposition of particles on the filter leads to changes in the mass of the system and results in changes of its frequency of oscillation. A microprocessor directly converts the vibration frequency to mass concentrations.

The flow rate through the analyser is controlled using thermal mass flow controllers and is automatically measured to determine the mass concentration. Air at $16.67 \, \mathrm{l \, min^{-1}}$ ($1.002 \, \mathrm{m^3 \, h^{-1}}$) is sampled through the sampling head and divided between the filter flow ($3 \, \mathrm{l \, min^{-1}}$) and an auxiliary flow ($13.67 \, \mathrm{l \, min^{-1}}$). The inlet is heated to $50 °C$ prior to particles being deposited onto the filter in order to eliminate the effect of condensation or evaporation of particle water. An identical impactor is used as a size-selective inlet for $PM_{10}$ on both instruments. For $PM_{2.5}$, the Partisol and the TEOM after 23 January 2002 were fitted with a Sharp-Cut Cyclone inlet; prior to that date the TEOM was fitted with a URG $PM_{2.5}$ cyclone inlet.

**2.2.2. Partisol Plus Model 2025**

The Partisol Plus Model 2025 from Rupprecht and Patashnick Co., Inc. is a microprocessor controlled measuring device for the sampling of particulate matter. Two Partisol instruments were installed on the Harwell site in order to measure simultaneously the $PM_{10}$ and $PM_{2.5}$ particles on a daily basis. These instruments use the same size-selective inlet designs as the ones used on the TEOM with the same flow rate of $16.67 \, \mathrm{l \, min^{-1}}$ ($1.002 \, \mathrm{m^3 \, h^{-1}}$). In contrast to the TEOM instrument, the inlet is not heated and air is sampled at ambient temperature and pressure.

The Partisol has a fully automatic filter exchange mechanism that provides unattended monitoring for up to 16 consecutive days for 24-h sampling prior to filter change. Temperature and pressure sensors are provided in the instrument with internal regulators to maintain the temperature within $±5 °C$ of ambient temperature. The total volume of air sampled, total measuring time, average temperature and pressure are recorded by the microprocessor. The Partisol Plus Model 2025 is equivalent to the gravimetric reference method according to the European standard prEN 12341 for the collection of $PM_{10}$ (see Mückler, 2000).

Quartz fibre filters (Whatman QMA 47 mm diameter filters, $0.6 \, \mu\mathrm{m}$ pore size) have been used for the collection of particulate matter. Pre-conditioning and post-conditioning of filters were undertaken in accordance with the requirements of prEN 12341. Blank and dust-loaded filters are handled according to the same protocols. Filters are equilibrated for 48 h within an air-conditioned weighing room at a temperature of $20 °C$ and a relative humidity of 50% before weighing on a balance with a resolution of $10 \, \mu\mathrm{g}$.

**2.3. Data for inclusion in the comparison**

The TEOM concentrations as reported by the instrument include a built-in calibration factor of $1.03 * \text{'TEOM reading'} + 3 \, \mu\mathrm{g}$. This calibration factor has been determined through intercomparison of data from TEOMs and filter-based reference instruments sampling a standard Arizona road dust in order to achieve US EPA certification (Patashnick and Rupprecht, 1991). The values corrected by the US EPA calibration factor correspond to data as supplied by the UK network (AURN data).

Examination of the influence of this calibration factor on the relationships between TEOM and Partisol $PM_{10}$ data for seven sites in the UK has shown that the calibration factor applied to TEOM values explains a large component of intercepts significantly higher than zero (results not shown). These intercepts have no physical meaning. On the contrary, linear models computed with raw TEOM readings have intercepts close to zero (for the relationship of Partisol versus TEOM with $R^2 > 0.70$). In this paper, the raw TEOM readings (i.e. with the calibration factor removed) have been used because they correspond to the true mass collected with the TEOM instrument.

$PM_{10}$ and $PM_{2.5}$ have been measured at the Harwell site using two collocated TEOMs for a number of years. Gravimetric data from two Partisol Plus 2025 have been derived from samples from September 2000. Data from 30 September 2000 to 2 July 2002 for the $PM_{10}$ and from 7 September 2000 to 19 June 2002 for the $PM_{2.5}$ are included in the comparison. A total of 436 paired 24-h $PM_{10}$ data and 461 paired 24-h $PM_{2.5}$ data are available for this study.

Particulate nitrate data used in this paper have been derived from samples collected using a collocated Partisol Plus Model 2025 instrument with PTFE filters and measured by ion chromatography.

Meteorological data (temperature and dew point used to compute relative humidity data) have been provided by the British Atmospheric Data Centre (BADC).

**3. Results**

**3.1. Comparison of TEOM with gravimetric Partisol particle mass**

Table 1 presents the mean and median $PM_{10}$ and $PM_{2.5}$ particle mass concentrations collected with the filter-based Partisol samplers and TEOMs. Standard deviations and interquartile ranges are computed to give an estimation of how the particle mass concentrations are spread out. Robust estimators (median and interquartile range) are also presented because they are not influenced by occasional high concentrations like the mean and the standard deviation.

In agreement with other studies (Allen et al., 1997; Ayers et al., 1999; Soutar et al., 1999; APEG, 1999; Salter and Parsons, 1999; Williams and Bruckmann, 2001; Cyrys et al., 2001), the results show that the TEOMs give lower particle mass than the filter-based gravimetric methods. The median relative differences

Table 1
Average $PM_{10}$ and $PM_{2.5}$ concentrations and number of exceedence days for $PM_{10}$ for the period September 2000–July 2002 ($N = 436$ for $PM_{10}$; $N = 461$ for $PM_{2.5}$)

| | PM concentrations | | Ratios Partisol/TEOM |
|---|---|---|---|
| | Partisol | TEOM[a] | |
| *PM_10* | | | |
| Arithmetic mean $\pm$ SD | 18.1 ($\pm 10.2$) | 10.2 ($\pm 5.3$) | 1.8 ($\pm 0.51$) |
| Median $\pm$ IQR[b] | 15.7 ($\pm 9.9$) | 9.2 ($\pm 5.5$) | |
| Daily concentration $> 50 \,\mu g/m^3$ | 8 | 0 | |
| | | | |
| *PM_2.5* | | | |
| Arithmetic mean $\pm$ SD | 12.3 ($\pm 9.9$) | 6.5 ($\pm 4.2$) | 2.1 ($\pm 2.7$) |
| Median $\pm$ IQR | 9.0 ($\pm 8.8$) | 5.3 ($\pm 4.4$) | |

[a] TEOM data are raw mass with USEPA correction removed.
[b] IQR = interquartile range.

between Partisol and TEOM particle mass concentrations are, respectively, 42% for $PM_{10}$ and 48% for $PM_{2.5}$.

The median $PM_{10}$ and $PM_{2.5}$ concentrations show smaller differences between Partisol and TEOM data than the means, indicating that some high concentrations measured by the Partisol instruments contribute to the higher difference in the means. The higher standard deviations (or interquartile ranges) for Partisol data than for TEOM data lead to the same conclusion. This means that the TEOMs read lower to a greater extent at high concentrations and suggests that higher amounts of semi-volatile compounds characterise events of high particulate matter concentrations.

The mean ratio Partisol/TEOM and the corresponding standard deviation for the whole studied period are presented in Table 1. Ratios of 1.8 for $PM_{10}$ data and of 2.1 for $PM_{2.5}$ data are found, both with high standard deviations, showing that a large range of ratios occurs. This result is in agreement with other studies (King et al., 2000; Cyrys et al., 2001; Green et al., 2001) showing the variability of the difference between measurements from TEOMs and gravimetric methods from one day to another.

The European Limit Value for particles measured as $PM_{10}$ is $50 \,\mu g\, m^{-3}$ (24 h mean) with up to 35 permitted exceedences per year. The number of days exceeding this daily standard for the September 2000–July 2002 period is also presented in Table 1. No exceedence days were measured by the TEOM, while the gravimetric Partisol method gives a total of 8 days exceeding $50 \,\mu g\, m^{-3}$. Most of these events corresponded to daily Partisol concentrations above $60 \,\mu g\, m^{-3}$, while the corresponding TEOM particle mass concentrations were generally below $30 \,\mu g\, m^{-3}$. Seven of these events corresponded to cold days (temperature below $10°C$) including four to very cold days (temperature below $5°C$) and three of these events corresponded to days with high relative humidity (above 90%). This suggests that semi-volatile compounds and occasionally particle-bound water on Partisol filters are responsible for large differences between instruments. For all these events, the $PM_{2.5}$ fraction represented more than 80% of the $PM_{10}$ fraction (and often more than 90%); while the average $PM_{2.5}/PM_{10}$ ratio for the studied period is 60%. This means that episodes of high concentrations of $PM_{10}$ concentrations at Harwell are driven by high $PM_{2.5}$ concentrations. The mean ratio Partisol/TEOM for these events of high concentrations is 2.1, which is higher than the mean ratio for the whole study period, confirming the greater difference at high concentrations.

The best fitting curves for the TEOM-Partisol relationships are linear despite the higher differences at high concentrations (see Fig. 1). Some studies have actually found nonlinear relationships due to higher amounts of volatile species at higher particle mass concentrations; resulting in higher divergences at higher concentrations (Salter and Parsons, 1999; APEG, 1999). Despite the scatter of the data, reasonable linear regressions between TEOMs and Partisol data can be established (Fig. 1). These linear regressions have been computed using the reduced major axis (RMA) regression method. Unlike the least square regression method that assumes that observations of the independent ($X$) variable are accurate, this method makes no assumption about the set of $X$-variable observations. A recent paper from Ayers (2001) has shown that the least square regression analysis is not appropriate for an instrument comparison exercise and leads to biased evaluations of the relationship between two instruments.

The relationships found for Harwell are not improved using nonlinear models and not really improved nor changed by removing outlier points. These results show that a large part of Harwell particle mass collected with the Partisol is vaporized in the inlet of the TEOMs. Almost all TEOM particle mass concentrations are lower than the Partisol particle mass concentrations.

[Figure]

Fig. 1. TEOM versus Partisol for (a) $PM_{10}$ and (b) $PM_{2.5}$ ($N = 436$ for $PM_{10}$; $N = 461$ for $PM_{2.5}$).

Possible explanations for these substantial underestimations are examined below.

**3.2. Origins of the divergence and influencing parameters**

The results suggest that some semi-volatile compounds are lost in the inlet of the TEOMs. The major inorganic semi-volatile compound is thought to be particulate ammonium nitrate. Its gas–particle partitioning depends on the atmospheric temperature and relative humidity (Stelson and Seinfeld, 1982a, b). Particulate ammonium chloride is another well-known semi-volatile inorganic compound (Pio and Harrison, 1987a, b). Except in specific polluted areas, the concentrations of ammonium chloride in the atmosphere are generally too low to lead to condensation, and particulate ammonium chloride is considered negligible in this study. A large range of semi-volatile organic compounds in small amounts but of non-negligible total mass are contained in particulate matter. The gas–particle partitioning of these organic semi-volatile compounds depends on a large number of parameters such as their concentration, the total particle surface area, the particle composition, the total particulate matter concentration, the atmospheric temperature and the relative humidity (US EPA, 1996). Semi-volatile compounds are expected to be mainly in the fine fraction of particles. Particulate ammonium nitrate and semi-volatile organic compounds are thought to be the main compounds lost in the inlet of the TEOM instrument.

Particle-bound water associated with particles collected with the Partisol constitutes another explanation for the difference between the two instruments. The equilibration of Partisol filters to a relative humidity of 50% does not remove all particle-bound water. The examination of the particle water content as a function of relative humidity shows a significant increase for humidity larger than 30% (US EPA, 1996). Due to hysteresis effects the mass of water is greater in samples collected at high humidity (as generally prevails in the UK) and transferred to a lower humidity environment. A recent study (Price et al., 2003) concluded that the retention of water by the filter of the gravimetric method

[Figure]

Fig. 2. Difference between Partisol and TEOM for $PM_{10}$ data versus difference between Partisol and TEOM for $PM_{2.5}$ data ($N = 370$)—data in $\mu g\,m^{-3}$.

was the major cause of differences with the TEOM for their site in Northern England.

Finally, adsorption of organic vapour by the Quartz filters used in the Partisol instruments may contribute to the difference between the two instruments. Quartz fibre filters have a large specific surface area upon which adsorption of gases can occur. A large positive artefact due to adsorption of organic gases onto Quartz fibre filters is now well-known (Turpin et al., 2000). The importance of this artefact associated with the sampling of particulate matter with Quartz fibre filters cannot be estimated in this study. The contribution of particulate ammonium nitrate and the influence of temperature and relative humidity are examined, since appropriate data were available.

The close agreement between the difference between Partisol and TEOM data for $PM_{10}$ and for $PM_{2.5}$ shows that most of the material not measured by the TEOMs is contained in the $PM_{2.5}$ fraction (see Fig. 2). The series of negative values of differences for $PM_{2.5}$ (i.e. TEOM data higher than Partisol data) are considered as suspicious. These values are not associated with negative values of differences for $PM_{10}$. They may be the result of artefacts with the $PM_{2.5}$ TEOM sampler. Allen et al. (1997) have observed large fluctuations in the $PM_{2.5}$ mass concentrations measured with the TEOM due to rapid changes in the composition of air masses (pollutant concentrations and ambient relative humidity) affecting the equilibrium of particles collected on the TEOM filter and then responsible for adsorption or desorption of particulate material. They did not observe the same phenomenon for the simultaneous $PM_{10}$ TEOM measurements and assumed that the coarse fraction of particles stabilises the semi-volatile particulate matter collected simultaneously. If these suspicious values are removed, a close linear relationship of $y = 0.973x + 2.167$, $r^2 = 0.78$ is found between the difference for $PM_{10}$ and difference for $PM_{2.5}$. A slope close to 1 with a small systematic difference of about $2\,\mu g\,m^{-3}$ (intercept) is shown by this relationship. About the same difference is found comparing the average differences between Partisol and TEOM data for $PM_{10}$ and $PM_{2.5}$ (median of differences for $PM_{2.5}$, $4.0\,\mu g\,m^{-3}$; for $PM_{10}$, $6.3\,\mu g\,m^{-3}$). This suggests that a small part of the difference between the two methods is in general associated with the $PM_{coarse}$ fraction. Nevertheless, a few much larger differences between Partisol and TEOM data for $PM_{10}$ than for $PM_{2.5}$ are observed for low values of differences for $PM_{2.5}$. These much larger differences (above $15\,\mu g\,m^{-3}$) occurred during damper days (relative humidity above 90%) and might be the result of particle-bound water associated with $PM_{coarse}$ particles (e.g. a major compound, NaCl is very hygroscopic and is mainly found in the coarse fraction of particulate matter).

The particulate ammonium nitrate mass was computed from the particulate nitrate assuming that all the particulate nitrate is associated with ammonium ions. In practice involatile nitrates (e.g. sodium nitrate) may make an important contribution to airborne nitrates. The particulate ammonium nitrate is compared to the difference between the two instruments and also used to correct the TEOM particle mass concentrations. Fig. 3 represents the relationships between the sums of TEOM particle mass data and calculated particulate ammonium nitrate with Partisol particle mass data; and in Fig. 4, the relationships between the difference between Partisol and TEOM data with the calculated particulate ammonium nitrate are plotted.

Both $PM_{10}$ and $PM_{2.5}$ TEOM data are significantly improved by adding the particulate ammonium nitrate. The slopes of the relationships with the Partisol data are closer to the 1:1 line and the correlation coefficients are better showing that the data are less scattered. For the

[Figure]

Fig. 3. TEOM PM data (empty dots) and TEOM PM data + particulate ammonium nitrate (full dots) versus Partisol PM data: (a) $PM_{10}$ ($N = 146$) and (b) $PM_{2.5}$ ($N = 161$).

[Figure]

Fig. 4. Difference between Partisol and TEOM PM data versus calculated ammonium nitrate: (a) $PM_{10}$ ($N = 146$) and (b) $PM_{2.5}$ ($N = 161$)—data in $\mu g\,m^{-3}$.

concentrations of particulate ammonium nitrate higher than $2–3\,\mu g\,m^{-3}$, the relationships between the difference between Partisol and TEOM data and the particulate ammonium nitrate are fairly linear (see Fig. 4); while for the concentrations lower than $2–3\,\mu g\,m^{-3}$, the relationship is poor indicating a significant relative contribution of other species. The remaining differences between Partisol and TEOM data are on average about 2.7 and $4.9\,\mu g\,m^{-3}$, respectively, for $PM_{2.5}$ and $PM_{10}$ and seem to be constant for the higher differences. Particulate ammonium nitrate (median concentration about $1.9\,\mu g\,m^{-3}$, during the study period) is lost in the inlet of the TEOMs and contributes substantially to the lower reading of the TEOM, but its volatilisation does not explain the whole difference between Partisol and TEOM data. It corresponded on average to about 26% of the material lost in the $PM_{10}$ fraction and about 40% of the material lost in the $PM_{2.5}$ fraction at Harwell. Different results in other studies reflect the different aerosol composition at different sites and explain the wide range of relationships between TEOM and gravimetric reference method found in the literature. Allen et al. (1997) found that the entire difference between a TEOM and a manual filter-based method for the Rubidoux site (California) can be attributed to particulate ammonium nitrate. On the contrary, Cyrys et al. (2001) have a small contribution of ammonium nitrate at their site. Harwell lies between these two extremes.

At the Harwell site, particulate ammonium nitrate represents an important part of the particulate material not measured with the TEOM instruments during episodes of high particulate pollution. Particulate nitrate data are available for three out of the eight episodes for which the gravimetric $PM_{10}$ particle mass exceeded $50\,\mu g\,m^{-3}$. These three events were all associated with a high concentration of particulate nitrate (equivalent $NH_4NO_3$ concentrations above $16\,\mu g\,m^{-3}$) and the particulate ammonium nitrate represented more than 50% of the particulate material lost. For two of them, the TEOM data are successfully corrected by adding the particulate ammonium nitrate; while for the last one, the loss of the particulate ammonium nitrate does not explain the entire difference between the two instruments (about 18% is still uncorrected). The five other episodes with $PM_{10} > 50\,\mu g\,m^{-3}$ measured with the gravimetric Partisol method and not measured with the TEOM have ratios Partisol/TEOM above the mean ratio Partisol/TEOM and occurred during cold days, indicating that high amounts of semi-volatile compounds were lost. These results suggest that episodes dominated by high secondary particulate pollutants are not measured by the TEOM at the Harwell site.

The amount of semi-volatile compounds associated with the particles is expected to depend on the temperature and the relative humidity. Additionally, the relative humidity has a significant impact on the growth of particulate matter. Comparisons between TEOMs and reference gravimetric methods in different countries have shown that for warmer and dryer regions the agreement is better than for colder and damper regions (Williams and Bruckmann, 2001; Noack et al., 2001). Similarly, other studies have shown that the agreement is better during the warmer months of the year than during the colder months (Allen et al., 1997; Williams and Bruckmann, 2001). Price et al. (2003) reveal that the divergence at their sampling site is higher on damper and/or rainy days. All these studies seem to confirm the influence of these meteorological parameters, but a previous study (Cyrys et al., 2001) did not find any relationship between the underestimation of the TEOM and the temperature or the relative humidity.

Because of the seasonal variation of the underestimation of the TEOM, Williams and Bruckmann (2001) recommend in their "Guidance to member states on PM monitoring and intercomparisons with the reference method" examination of the seasonal variations of factors and equations to amend the TEOM data. This examination for the Harwell site is not very helpful because of the small contrast between the seasons (ratio Partisol/TEOM for summer, 1.73; for winter, 1.88).

The relative differences (differences between the gravimetric Partisol mass and the TEOM mass divided by the Partisol mass) versus the temperature and versus the relative humidity are presented, respectively, in Figs. 5a and b. These results confirm that the differences between the TEOM and the Partisol for $PM_{10}$ mass concentrations measured at Harwell depend on both the temperature and the relative humidity. The difference between the two instruments decreases with the temperature and increases with relative humidity. However, a large range of relative differences is associated with each temperature and relative humidity bin, suggesting a simultaneous influence of the particle composition. The same results are found for the $PM_{2.5}$ at Harwell and for other sites in the UK (results not shown).

Fig. 5a clearly shows that the differences between Partisol and TEOM data are higher than 40% for lower temperatures (below $2^\circ C$) and lower than 25% for higher temperatures (above $18^\circ C$). Fig. 5b shows that half of the differences between Partisol and TEOM data are below 30% for relative humidity lower than 70% and higher than 50% for relative humidity higher than 90%. The examination of the ratio Partisol/TEOM with both temperature and relative humidity leads to similar conclusions. The mean ratio for colder and damper weather ($T < 10^\circ C$ and $RH > 80\%$) is 1.98 and for warmer and drier weather ($T > 10^\circ C$ and $RH < 80\%$) is 1.63.

The decrease of the difference between the TEOM and the Partisol with temperature is likely to be related to a decrease in the amount of semi-volatile compounds in

[Figure]

Fig. 5. Box plots of relative differences between Partisol and TEOM $PM_{10}$ data for different (a) temperature ranges and (b) relative humidity ranges.

the particles or to an increase of the semi-volatile material volatilised from the Partisol filters, while the increase with relative humidity is likely to be related to the increase of the water content of particles and possibly also to the increase of semi-volatile compounds in particulate matter (as an example, the equilibrium of the ammonium nitrate between gaseous and condensed phases depends on the relative humidity, see Stelson and Seinfeld, 1982a, b). The influence of relative humidity on the difference between TEOM and gravimetric data seems to be less important than the influence of the temperature. The larger range of percentages associated with each range of humidity might be the result of the variation of particle composition. It might also be the result of the hysteresis of some major particulate compounds (e.g. $(NH_4)_2SO_4$) responsible for complicated relationships between particles and water (Speer et al., 2003). The significant influence of the temperature suggests that the lower reading of the TEOM is mainly due to the loss of semi-volatile compounds in the TEOM inlet.

**4. Conclusion**

An intercomparison between two TEOMs and two Partisol samplers for the monitoring of particles as $PM_{10}$ and $PM_{2.5}$ has been carried out at a rural site located in Oxfordshire, UK. Results confirm the expected large difference between TEOM reading and particle mass from filter-based gravimetric methods. On average almost half of the particle mass derived from the Partisol instruments is not measured by the TEOMs. Because most of the particulate material not measured by the TEOMs is in the $PM_{2.5}$ fraction of particulate matter, the relative difference between the two methods is larger for $PM_{2.5}$ than for $PM_{10}$. Strong day-to-day variations of the difference between TEOM and Partisol particle mass are observed. This variability leads to difficulty in establishing statistical relationships that successfully amend TEOM data.

Results suggest that the major semi-volatile compound lost in the inlet of the TEOMs is ammonium nitrate. Adding the calculated ammonium nitrate mass to the TEOM mass significantly improves the mass estimate, accounting for about 40% of the $PM_{2.5}$ mass deficit. Loss of other particulate material such as semi-volatile organic compounds and water must also contribute since the loss of particulate ammonium nitrate does not explain the entire difference. Large differences between measurements are associated with high particulate ammonium nitrate concentrations. As a consequence, episodes of high particulate matter concentration dominated by secondary pollutants at Harwell are not measured by the TEOMs.

The influences of temperature and of relative humidity on the difference between TEOMs and Partisol particle mass have been demonstrated. Elevated temperatures or lower relative humidities reduce the divergence between measurements. This is in agreement with previous studies showing that in summer or in areas with warm and dry climates, the agreement between TEOMs and reference gravimetric methods is better than in winter or in areas with colder and damper climates. These influences are consistent with the material lost in the inlet of the TEOMs being semi-volatile compounds and particle-bound water. The loss of semi-volatile substances from the Partisol filters reducing the divergence between the two methods during elevated temperatures is not precluded.

The establishment of a model to correct the TEOM particle mass by including influential variables such as particle composition, temperature and relative humidity would be a better alternative than statistical models. However, this study suggests that semi-volatile organic compounds may constitute a significant part of the particulate material lost in the TEOMs. This component of particulate matter includes a large range of compounds that cannot be measured on a daily basis and moreover, the physico-chemical properties of the vast majority of such compounds are poorly quantified. On the other hand, artefacts from filter-based gravimetric methods possibly contribute to the difference and should be quantified.

**Acknowledgements**

This project was funded by the Department for Environment, Food and Rural affairs (Contract No. EPG 1/3/184). Meteorological data are supplied by the British Atmospheric Data Centre (BADC).